# Edible Alginate–Lecithin Films Enriched with Different Coffee Bean Extracts: Formulation, Non-Cytotoxic, Anti-Inflammatory and Antimicrobial Properties

**DOI:** 10.3390/ijms252212093

**Published:** 2024-11-11

**Authors:** Robert Socha, Aleksandra Such, Anna Wisła-Świder, Lesław Juszczak, Ewelina Nowak, Karol Bulski, Krzysztof Frączek, Ivo Doskocil, Barbora Lampova, Aneta Koronowicz

**Affiliations:** 1Department of Food Analysis and Evaluation of Food Quality, Faculty of Food Technology, University of Agriculture in Krakow, ul. Balicka 122, 30-149 Kraków, Poland; robert.socha@urk.edu.pl (R.S.); rrjuszcz@cyf-kr.edu.pl (L.J.); 2Department of Human Nutrition and Dietetics, Faculty of Food Technology, University of Agriculture in Krakow, ul. Balicka 122, 30-149 Kraków, Poland; 3Department of Chemistry, Faculty of Food Technology, University of Agriculture in Krakow, ul. Balicka 122, 30-149 Kraków, Poland; anna.wisla-swider@urk.edu.pl (A.W.-Ś.); ewelina.nowak@urk.edu.pl (E.N.); 4Department of Dietetics and Food Studies, Faculty of Science and Technology, Jan Długosz University in Częstochowa, Al. Armii Krajowej 13/15, 42-200 Częstochowa, Poland; 5Department of Microbiology and Biomonitoring, Faculty of Agriculture and Economics, University of Agriculture in Krakow, Al. Mickiewicza 21, 31-120 Kraków, Poland; karol.bulski@urk.edu.pl (K.B.); krzysztof.fraczek@urk.edu.pl (K.F.); 6Department of Microbiology, Nutrition and Dietetics, Faculty of Agrobiology, Food and Natural Resources, Czech University of Life Sciences Prague, Kamycka 129, 165 21 Praha, Czech Republic; doskocil@af.czu.cz (I.D.); lampova@af.czu.cz (B.L.)

**Keywords:** biopolymer-based packaging, sodium alginate, lecithin, coffee extracts, antioxidant activity, cytotoxicity, nitric oxide, antimicrobial activity, edible packaging

## Abstract

The aim of this study was to analyze the functional properties of newly obtained films based on sodium alginate and lecithin with the addition of antioxidant-rich coffee extracts and to verify their potential as safe edible food packaging materials. In our study, we developed alginate–lecithin films enriched with green or roasted coffee bean extracts. The roasting process of coffee beans had a significant impact on the total phenolic content (TPC) in the studied extracts. The highest value of TPC (2697.2 mg GAE/dm^3^), as well as antioxidant activity (AA) (17.6 mM T/dm^3^), was observed for the extract of light-roasted coffee beans. Films with the addition of medium-roasted coffee extracts and baseline films had the highest tensile strength (21.21 ± 0.73 N). The addition of coffee extract improved the barrier properties of the films against UV light with a decrease in the transmittance values (200–400 nm), regardless of the type of extract added. Studies on Caco-2, HepG2 and BJ cells showed that digestated films were non-cytotoxic materials (100–0.1 μg/cm^3^) and had no negative effect on cell viability; an increase was noted for all cell lines, the highest after 48 h in a dose of 1 μg/cm^3^ for a film with medium-roasted coffee (194.43 ± 38.30) for Caco-2. The tested films at 20% digestate concentrations demonstrated the ability to reduce nitric oxide (NO) production in the RAW264.7 cell line by 25 to 60% compared to the control. Each of the tested films with coffee extracts had growth inhibitory properties towards selected species of bacteria.

## 1. Introduction

Edible films and coatings can be produced from natural polymers, such as proteins, polysaccharides and lipids or their combination, which are perfectly biodegradable and safe for the environment [1]. However, the main disadvantages of lipid-based films and coatings are their opaqueness, fragility and instability [2]. Among the edible films based on hydrocolloids, alginate films have attracted particular interest for maintaining quality and extending the shelf-life of fruits, vegetables, meat, poultry, seafood and cheeses by reducing dehydration, controlling respiration, enhancing product appearance and improving mechanical properties [2].

Alginate itself is characterized by unique colloidal properties, including thickening, film and gel formation and emulsion stabilization, whereas alginate films are strong and resistant to oil and grease [3]. On the other hand, they do have not good water resistance due to their high hydrophilicity [4]. Therefore, this kind of polysaccharide is generally mixed with other biopolymers in order to improve its mechanical properties [4]. Alginate-based films used as packaging materials can be divided into two groups: The first one made only from alginate has poor water resistance. In the case of the second one, made from alginate and other biopolymers, the limitation has not been mentioned [5,6,7,8]. Complex films containing other polymers besides alginate are characterized by better mechanical properties. For example, the new polysaccharide film packaging prepared from alginate and citrus pectin exhibits good tensile strength and elongation at break [3]. What is more, the addition of antioxidants or extracts rich in antioxidants to alginate-based films results in better water and oxidant resistance [9,10]. The alginate-based film enriched with vitamin C could be stored in the dark at refrigeration for up to five months [11]. Since alginate is widely found in seaweed, a low cost is also the main advantage [4].

The main advantage of the application of an edible coating with incorporated antioxidants is the decrease in oxidation due to the gas barrier properties of the alginate coating and the synergistic effect between these two factors [2].

Surfactants are key ingredients used to improve the adhesion of coating materials [12]. Soy lecithin, as a surfactant and component of edible films or coatings, particularly impacts their color, solubility, opacity and microstructure [13]. Besides the factors mentioned above, lecithins may affect the antimicrobial properties of the films [14].

Coffee is one of the most widely consumed beverages in the world and is prepared from green or roasted coffee beans. The health-promoting properties of a coffee brew are to a great extent determined by the presence of the phenolic compounds in green coffee beans, as well as both the phenolic compounds and melanoidins in the case of roasted coffee beans. The predominant group of phenolic compounds present in coffee are chlorogenic acids [15].

Taking into account the bioactive properties of both green and roasted coffee beans, many in vitro and in vivo studies have shown the hypoglycemic, antiviral, hepatoprotective and immunoprotective activity of coffee extracts [15]. Moreover, consumption of coffee brews results in a decrease in 8-hydroxydeoxyguanoside, which is widely used as a biomarker of oxidative damage [16].

It was stated that some components of coffee brew, such as caffeine, volatile and non-volatile phenolics and phenolic acids, are reported to have an antimicrobial activity. Regarding the properties of phenolic compounds, Almedia et al. [17] have observed that of all the bioactive compounds of coffee brews, caffeine, trigoneline and protocatechuic acid, are the most potential antimicrobial agents.

Studies on the in vitro safety of coffee (in the form of aqueous extracts) have shown the influence of the degree of roasting of coffee beans on their cytotoxic effect in human cells [18]. Da Silva et al. in a cytotoxicity assay of water-soluble tetrazolium salt-1 (WST-1) on the human hepatocellular carcinoma cell line (HepG2) showed that in the case of commercial quality samples, cancer cells were more sensitive to the cytotoxic effect of the extract than the dark-roasted sample (IC_50_ 0.2244 mg/mL), followed by lightly roasted samples (IC_50_ 0.3721 mg/mL) and the medium-roasted sample (IC_50_ 0.4343 mg/mL) [18].

Among the special quality samples, it was found that HepG2 cells were more sensitive to treatment with the medium-roasted sample, and the percentage of cell death at the lowest concentration reached 76.58 ± 2.99%. For the light-roasted sample, the IC_50_ was 0.1849 mg/mL, followed by 0.2927 mg/mL for the dark-roasted sample [18].

The cell proliferation study showed that a 2 h incubation with green coffee bean extracts (GCBE) (homogenized powdered samples dissolved in 100 mg/mL stock solutions in 1% (*v*/*v*) DMSO) did not affect the proliferation of adenocarcinomic human alveolar basal epithelial cells (A549) and cancer cells of human esophagus (OE-33) in concentrations of 10–1000 μg/cm^3^, while it reduced proliferation in a human epithelial cell line derived from a colon carcinoma (Caco-2) and T24 bladder cells after incubation with the highest concentration (1000 μg/mL) [19].

The aim of this study was to obtain new alginate–lecithin films with the addition of coffee extracts from beans of different degrees of roasting. We hypothesized that the obtained films would be materials with good functional properties that determine their suitability as edible food packaging. We aimed to exclude cytotoxicity against human cells, determine the anti-inflammatory, antimicrobial and physicochemical properties of the tested films and investigate the phenolic profile and antioxidant activity of the films.

## 2. Results and Discussion

### 2.1. Total Phenolic Content (TPC) of the Coffee Extracts

The values of the TPC for the studied extract, expressed as mg of GAE per 1 dm^3^ of extract, are collected in Table 1. It was stated that the roasting process of coffee beans had a significant impact on the TPC in the studied extracts (Table 1). The highest value of this parameter (2697.2 mg GAE/dm^3^) was observed for light-roasted coffee beans, whereas the case of dark-roasted ones had the lowest (2057.8 mg GAE/dm^3^) (Table 1). With respect to the coffee beans, the most important components that can react with Folin–Ciocalteu’s reagent may be divided into two groups. The first one includes coffee phenolics, mainly chlorogenic acids. This class of compounds, antioxidant in character, is present in significant quantities in green coffee beans and in smaller quantities in roasted ones, whereas the second group includes melanoidins, whose content increases with increasing degrees of roasting, and which are present only in roasted coffee beans [20]. It was found that the TPC in light-roasted coffee is significantly higher than that observed in green coffee (2267.2 mg GAE/dm^3^). During the roasting of green coffee beans, the content of chlorogenic acids decreases, resulting in the formation of melanoidins [21]. Similarly to phenolic acids, these latter compounds are also characterized by strong antioxidant activity [22]. Thus, a decrease in phenolic acid content during roasting may be partially compensated by an increase in melanoidin content, which resulted in an increase in TPC value for an extract of lightly roasted coffee when compared with that of green ones. The balance between natural coffee phenolics and melanoidin content is the most favorable in the case of light-roasted coffee beans.

### 2.2. Antioxidant Activity of the Coffee Extract

The values of antioxidant activity of the coffee extracts, expressed in mM of Trolox (T)/dm^3^, were collected in Table 1. The results showed that all the extracts had a significant AA, but their values fluctuated considerably. Similarly as in the TPC, the light-roasted coffee exhibited the highest AA (17.6 mM T/dm^3^), followed by the medium-roasted sample (16.4 mM T/dm^3^), whereas the dark-roasted coffee extract was the lowest (12.8 mM T/dm^3^). The green coffee sample was characterized by an average value (14.6 mM T/dm^3^) among all the extracts. Similarly to the TPC, the AA of light and medium coffee exceeded that of the green one despite the fact that the last sample was the richest in natural phenolic (chlorogenic acids). Thus, the obtained results confirm a high AA of melanoidins present in roasted coffee beans. In our study, a very high linear correlation (r = 0.9985) was observed between the AA and TPC for the coffee extracts. However, besides chlorogenic acids and melanoidins, other factors are also involved in creating antioxidant properties, including all the non-phenolic compounds generated in all stages of the Maillard reaction undergoing during roasting process. With respect to the aforementioned factors, the most important are those compounds generated during pyrolysis, in particular the volatile compounds soluble in water after brewing, namely guaiacol, 4-ethylguaiacol and 4-vinylguaiacol created after the composition of coffee phenolics [23]. Furthermore, a considerable part of organic sulfur compounds, such as furfurylthiol and the other thiols, containing a SH- group of strong antioxidant properties are present in roasted coffee as, e.g., methanethiol, 2-furfurylthiol, 3-methyl-3-furanthiol, 3-methyl-2-butene-1-thiol and many others [23,24]. Among the volatile compounds of antioxidant properties, vanillin is also present in both roasted coffee beans and brew, and its content increases strongly during the roasting process [25]. Also, unidentified constituents, including pigments, bitter compounds, etc., may be responsible for shaping the antioxidant activity of green and roasted coffee brew [23].

### 2.3. Chlorogenic Acid (CGA) Content in the Coffee Extracts

The amounts of individual chlorogenic acids, i.e., chlorogenic (5-CQA), cryptochlorogenic (4-CQA), neochlorogenic (3-CQA), 3,4-dicaffeoylquinic (3,4-diCQA), 3,5-dicaffeoylquinic (3,5-diCQA) and 4,5-dicaffeoylquninic (4,5-diCQA) acids, in the coffee extracts, expressed as mg per 1 dm^3^, are specified in Table 1. The total content of CGAs ranged from 405 to 3208 mg/dm^3^ for the dark-roasted and green coffee extract, respectively. Among the CGAs under study, 5-CQA was present at the highest content, ranging from 232.04 to 2052.66 mg/dm^3^ of coffee extract, comprising 57% to 64% of the total CGA content. On the contrary, 4,5-diCQA was present at the lowest level, ranging from 6.24 to 86.31 mg/dm^3^. CGA contents were detected in a decreasing order, 5-CQA > 4-CQA > 3-CQA > 3,5-diCQA > 3,4-CQA > 4,5-CQA. The different order in the case of dicaffeoylquinic acids (diCQAs) was observed by Fuijoka and Shibamoto [26] who investigated various brands of coffees, where diCQAs content was detected in a decreasing order 3,4-diCQA > 4,5-diCQA >3,5-diCQA. This fact may result from differences between the coffee samples in terms of their botanical and geographical origin. The roasting process of coffee beans resulted in a decrease in CGA content in the extracts. The extract of green coffee was characterized by the highest level of all the investigated CGAs; however, the obtained data did not correspond to the values of TPC and AA. Extract of light-roasted coffee was characterized by the highest values of the latter parameters. That phenomenon may result from the fact that during the light roasting of green beans, the process of thermal decomposition of chlorogenic acids was compensated by the formation of melanoidins exhibiting strong antioxidant properties [20]. Thus, the most optimal balance between chlorogenic acids and melanoidins (in terms of antioxidant activity) was achieved for light-roasted coffee.

### 2.4. Free Phenolic Acid (FPA) Content in the Coffee Extracts Determined After Alkaline Hydrolysis

Despite many reports focusing on the presence of chlorogenic acids in coffee, scarce are the data concerning research on the content of the free phenolic acids (i.e., caffeic, *p*-coumaric and ferulic acid) creating the structure of chlorogenic acids [27]. Thus, in order to determine FPA content, an alkaline hydrolysis was performed and resulted in a release of FPAs from the bounded forms. The amounts of individual FPAs (i.e., caffeic, *p*-coumaric and ferulic acids), expressed in mg/dm^3^ of extract, are specified in Table 2. Among the acids under study, caffeic acid was a predominant compound in all extracts, and its level ranged from 294.2 to 1594.74 mg/dm^3^. The presence of CA after alkaline hydrolysis was due to the hydrolytic decomposition of mono- and dicaffeoylquinic acids [24]. Ferulic acid turned out to be the second in terms of content acid and was formed as a result of hydrolytic decomposition of 3-feruolylquinic, 4-feruoylquinic and 5-feruoylquinic acids belonging to the group of chlorogenic acids. The level of *p*-coumaric acid ranged from 9.19 to 25.33 mg/dm^3^ of extract and was the lowest among the acids tested. The chlorogenic acids were sensitive to the roasting process, and the total level of CA in dark-roasted coffee (Table 2) comprised only 18.44% (in the form of CGAs) of its value for the green coffee extract. Similarly, as in the case of chlorogenic acids, green coffee extract was the richest in terms of FPAs content, whereas the dark-roasted coffee extract was the poorest. Again, the levels of FPAs did not correspond to the values of TPC and AA.

Statistically significant linear correlations (α = 0.05) were observed between the content of caffeic acid (CA) after hydrolysis and the contents of the following chlorogenic acids: 5-CQA, 4-CQA and 3-CQA, 3,4- and 4,5-di-CQA, as well as total CGAs (Table 3), which confirms the fact that caffeoylquinic acids as the bound forms of CA are the only precursors of phenolic acids in green and roasted coffee beans. With respect to *p*-coumaric and ferulic acids content, these values correlated significantly with 5-CQA, 3,4-, 3,5- and 4,5-di-CQA, as well as total CGAs (Table 3), which means that there is also a significant linear correlation between the contents of various hydroxycinnamic acids (i.e., CA, *p*-COA, FA) present in coffee beans. On the other hand, positive but not statistically significant correlations were found between TPC and AA determined spectrophotometrically and the content of both the free and bound (chlorogenic acids). This phenomenon can be explained by the fact that besides chlorogenic acids, as well as melanoidins (in the case of roasted coffee extracts), the other compounds may also be responsible for the shaping of the AA in the coffee extracts. Among these compounds, the most important are tocopherols (together with tocotrienols), free amino acids, unsaturated fatty acids, sulfur volatile compounds, diterpens and their derivatives and the other unidentified compounds [28].

### 2.5. ATR-FTIR Spectrophotometry

Fourier-transformed infrared (FT-IR) spectroscopy was used to identify the functional groups and was carried out to observe the structural interactions of alginate–lecithin films enriched with coffee extracts. Figure 1A,B displays six FTIR spectra of the films with (and without) the addition of the coffee extracts.

There was a very limited number of changes between the obtained absorption spectra (Figure 1A,B). A comparative evaluation of the FTIR bands indicated that most of the bands were similar in their band characteristics such as peak position and broad nature.

The FTIR spectra for alginic acid showed a typical infrared spectrum described in the literature [29,30,31]. The alginate film spectrum revealed a broad band at 3300 cm^−1^ for the O–H stretching and two bands at 1599 cm^−1^ and 1407 cm^−1^ characteristic for the asymmetric and symmetric stretching vibrations of carboxylate salt ions, respectively [29]. The band with the highest absorbance, in the range between 1100–950 cm^−1^ and at 1025 cm^−1^, is associated with C–C stretching of D-guluronic acid from alginate [30]. Further, the weak signal showing the presence of C–H stretching vibration [31] was observed in the spectra at 2849 cm^−1^.

In the present study, for the FTIR spectra alginate/lecithin films, the most intense bands are those corresponding to alginate. In the spectral region between 1100 and 1000 cm^−1^ and centered around 1027 cm^−1^, there are the overlapped PO_2_^−^ and P–O–C infrared active vibrations characteristic of lecithin lipids [32]. It is worth noting that lecithin has also a N(CH3)3 group band visible in the spectra at 920 cm^−1^ [33]. The broad peak centered at around 3400 cm^−1^ corresponds to the hydroxyl group from alcoholic esters. The weak band at 1737 cm^−1^ represents C=O esters groups, indicating phosphatidylethanolamine and phosphatidylserine moiety presented in lecithin [34].

The absorption spectra of the alginate/lecithin films with the coffee extracts show some bands of the coffee components like caffeine, lipids, chlorogenic acids and carbohydrates. Analysis of these spectra reveals important regions (1750–1650 cm^–1^ and 1200–1100 cm^–1^) that could be helpful for the comparison. The spectra show also some common and significant features, such as the bands at around 1121 cm^−1^ related to PO_2_^−^ and P–O–C infrared active vibrations characteristic for lecithins (especially for phosphatidylethanolamine and phosphatidylserine moiety). In addition, the band around 1737 cm^−1^ is the most visible in the films with roasted coffee extracts which may be associated with the compounds that are polyphenolic in character. The other bands that appear at 2914 and 2849 cm^–1^ correspond to the antisymmetric and symmetric vibrations of the methyl group (–CH3) [35], particularly those originating from lipids [36]. The peaks attributed to caffeine present at the wavenumber around 2849 cm^−1^ and in the region of 1650–1600 cm^−1^ [37] were overlapped with the peak of alginate carboxyl groups. The spectra region in the range 1800–1680 cm^−1^ is characteristic of the carbonyl group, especially of coffee ingredients like aromatic and aliphatic acids, ketones, aldehydes or aliphatic esters [38].

Nevertheless, a weak band at around 1740 cm^−1^ is attributed to the carbonyl (C=O) vibration associated with the ester group in triglycerides. Lyman et al. [38] also attributed the band in that region to aliphatic esters. The coffee lipid band was attributed also to the wavenumber centered around 2930 cm^−1^ that is related to the stretching vibration of the carbonyl and stretching asymmetric C–H of methylene groups. The band of chlorogenic acid present in the range of 1450–1400 cm^−1^ and centered around 1410 cm^−1^ [38] is better visible in the case of films enriched with medium- and light-roasted coffee extracts.

In the study concerning the chemometric analysis of processed Brazilian coffees, Ribeiro et al. [39] attributed the spectra in the range of 1700–1600 cm^−1^ to chlorogenic acids and caffeine concentration. Another author demonstrated that the caffeine band is commonly detected in the range between 1650 and 1600 cm^−1^ [40]. In our study, we cannot observe the single bands corresponding to caffeine and chlorogenic acids due to the strong absorption of the –COO- group (originating from alginate). However, the band stretch suggests that the spectra of other compounds are overlapped under the carboxylate band.

In the spectra corresponding to the films enriched with coffee extracts, the band corresponding to the –OH vibration was shifted to 3400 cm^−1^ and is broader than that for the alginate films. This phenomenon may result from the presence of phenols and carboxylic acids originating from the coffee extracts and the crosslinking interaction between these compounds and the alginate/lecithin film.

### 2.6. High-Performance Size Exclusion Chromatography (HPSEC–MALLS–RI)

High-performance size exclusion chromatography (HPSEC) coupled with multi-angle laser light-scattering (MALLS) and a differential refractive index (RI) detector was used to determine the weight-average molecular weight (M_w_) and radii of gyration (R_g_) of the alginate/lecithin films with coffee extracts (Table 4).

The results of the average molecular weights (Peak I and Peak 2) presented in Table 4 indicate that two fractions with different molecular weights can be determined in sodium alginate. This means that its structure is heterogeneous. The R_g_ values also suggest that the sizes of chain fractions differ significantly. The use of lecithin causes a large decrease in the values of these masses (M_w_) and the radii of gyration (R_g_), thus organizing the structure of the entire macromolecule.

The addition of coffee extracts caused a sharp increase in the molecular weight of the films, simultaneously causing a further decrease in the R_g_ value, which may indicate a lower polydispersity of the chains of the films thus prepared. The significant increase in molecular weight may be related to the interactions between the alginate–lecithin chains and the compounds contained in the coffee extracts.

The differential weight fraction vs. molecular weight plots for the samples of alginate, alginate/lecithin and alginate/lecithin films with the coffee extracts are illustrated in Figure 2.

A differential weight fraction is the mass share of a sample containing molecules with a molecular mass constituting a given fraction in relation to the total mass of the sample [41].

The pure alginate sample exhibited a much broader molecular weight distribution than the alginate/lecithin film and the films made with the addition of the coffee extracts. We observed that the polydispersity of pure alginate was higher than that of all the films tested. The addition of lecithin and coffee extracts causes a reduction in the molar mass polydispersity.

Figure 2 shows that sodium alginate is characterized by a wide band of molecular weight distribution, while the alginate–lecithin film as well as the films made with the addition of coffee extracts are characterized by an increase in the homogeneity of the sample.

The addition of various coffee extracts to alginate/lecithin films led to a significant increase in the molecular weight of eluted polysaccharide chains and exhibited a narrow M_w_ (Table 4) distribution. Concurrently, a significant decrease in the polydispersity of the eluted molecules was observed when compared with the films without the addition of the coffee extract.

It is worth noting that the addition of the coffee extracts changed the values of the weight-average molecular weights; however, it widely decreased the dispersity of the sample, which was reflected in the values of the radius of gyration. These significant increases in M_w_ (Table 4) could be caused by interactions (crosslinking) between the alginate/lecithin films and the phenolic compounds (chlorogenic acids) and melanoidins present in the coffee extracts. The phenolic acids can react with various components by covalent or non-covalent bonds. The covalent bonds include interactions with non-starch polysaccharides via ester bonds [42]. The observed increase in the intensity of the absorption band corresponding to the ester bond (around 1737 cm^−1^) may indicate the interaction of sodium alginate with phenolic compounds.

On the other side, the carboxyl and hydroxyl groups of phenolic acids are capable of binding to polysaccharides through hydrogen bonds, chelation or covalent bonds, forming bridges or crosslinks [43].

The shift in the absorption band at 1464 cm^−1^ characteristic of N^+^(CH_3_)_3_ groups may indicate the interaction of lecithin with –OH groups derived from phenolic acids. These changes may indicate the occurrence of hydrogen bonds between these groups.

Another research study [44] indicates that alginates interact with phenols mainly through the formation of hydrogen bonds involving the hydroxyl groups of phenols and both the hydroxyl and carboxyl groups of alginates. The typical hydrogen bonding between –(OH)2 and O6 (belonging to two consecutive carbohydrate units) was observed for poly(G) chains; in the case of poly(MG) (recurrent mannuronate (M) and guluronate (G) units) units, the corresponding hydrogen bonding was formed involving –(OH)3 groups (belonging to G units) as donors and O6 or O5 (belonging to M units) as acceptors. The most common type of interaction between the phenolic compounds and the alginate chains was found to be hydrogen bonding [44]. This includes both the carboxyl and hydroxyl groups of carbohydrates [44,45].

### 2.7. Thermal (DSC) Analysis of the Alginate Films

Evaluation of the biopolymer film using DSC analysis is a valuable source of information on the interactions between the tested polymer and the additives used during film preparation. These interactions may be responsible for changing the thermal properties of biopolymers. In order to find the presence of an interaction between alginate and lecithin as well as between the tested polymer and the components in the coffee extract, DSC spectra of the tested films were made and analyzed (Table 5).

For all tested samples, only two peaks (corresponding to the phase transitions) were found on the DSC spectra. The first one in the temperature range of 166.9–186.6 °C can be attributed to the endothermic fusion associated with the melting of the polysaccharide [46,47], whereas the second one in the temperature range of 232.4–242.2 °C was related to the exothermic fusion resulting from the polymer degradation processes. The alginate decomposition process is caused by a partial decarboxylation of the protonated carboxyl groups of alginate following the oxidation reaction of the polyelectrolyte [48]. With respect to our results (Table 5), the highest differences between the values of the studied thermal parameters, i.e., the temperature of endothermic fusion (associated with a melting process), and the heat of this fusion were observed between the film obtained for the pure alginate and the alginate with the addition of lecithin (5% w/w). The incorporation of lecithin caused a decrease in the temperature of the melting point of the alginate film by about 20 °C accompanied by an increase in the heat of this fusion (from 76.96 to 174.53 J/g, respectively). On the contrary, the addition of roasted coffee bean extracts to the AL film resulted in a decrease in the melting point of the polymer; however, it is much lower than that observed for the film with only the lecithin addition. With respect to the enthalpy of fusion (ΔH_m_) for melting, the observed increase (more than two-fold) in that value was probably caused by the formation of strong electrostatic interactions (ion–ion type) between the quaternary ammonium groups of lecithin and the negatively charged alginate carboxyl groups, as well as hydroxyl groups (attributed to ion–dipole interactions). Within the samples under study incorporated with coffee extracts, no statistically significant differences were found between the values of the melting temperature (T_m_) and the enthalpy ΔH_m_ of that process (Table 5), with the exception of the ALD sample for which ΔH (81.45 J/g) was significantly lower. A significant decrease in values of enthalpy in films enriched with roasted coffee components when compared with AL film was observed. That effect may result from a weakening in the strength of the electrostatic interactions between the alginate and lecithin and the formation of competitive interactions between the positively charged quaternary choline moiety of lecithin and the negatively charged carboxylic and/or hydroxyl groups of chlorogenic acids and melanoidins present in the roasted coffee extracts. Values of the temperature of exothermic fusion (attributed to a polymer decarboxylation) ranged from 232.4 to 233.9 °C; however, they did not differ significantly within the analyzed samples (Table 5). The statistically significant differences were only observed between the values of enthalpy of exothermic fusion (ΔH_m_), which ranged from 144.1 to 350.63 J/g. The highest value of heat of the decarboxylation process was observed in the case of the alginate with lecithin addition (350.63 J/g), for which that value was two times higher when compared to pure alginate. This fact proves the importance and strength of electrostatic interactions between the carboxyl groups of alginate and the positively charged lecithin used as a surfactant. Within the samples with the addition of coffee extracts, significant differences were also noticed between the values of enthalpy of exothermic fusion ΔH_m_ (Table 5). The highest value of ΔH_m_ (235.53 J/g) was observed for the sample with the addition of green coffee extract, while the lowest (144.10 J/g) was for the sample enriched with light-roasted coffee.

### 2.8. Mechanical Properties of the Films

For the alginate/lecithin films, the effect of coffee extract on their mechanical properties was also analyzed (Table 5).

Tensile strength (TS) is defined as the maximum load that a material can withstand without fracture. Values of this parameter for the films ranged from 108.5 to 172.3 MPa. The ALM film and the control sample had the highest tensile strength, whereas the ALG film had the lowest tensile strength (Table 5). With the exception of the ALM film, the enriched films have lower values of tensile strength in relation to the baseline film (Table 5). An analogous trend was observed for the second parameter characterizing the strength of the films, i.e., the maximum breaking load (MBL), with values ranging from 13.78 N for the ALL to 21.21 N for the ALM film. The mechanical properties of alginate and lecithin films are based on the bonding forces in the polymer network, i.e., strong electrostatic interactions between positively charged quaternary ammonium groups of lecithin and the negatively charged carboxyl and hydroxyl groups of alginate. This was also reflected in a significant increase in enthalpy value (from 76.97 to 174.53 J·g^−1^) for the endothermic process (melting) of the alginate/lecithin film compared to the pure alginate (Table 5). The higher the enthalpy value ΔH (for melting the film), the more energy is required to break the interaction between lecithin and alginate. Ismallayli et al. observed that the carbonyl group of the alginate and ammonium groups of chitosans can have electrostatic interactions in the case of alginate/chitosans films [6]. Bioactive antioxidants present in coffee extracts with different roasting levels, such as chlorogenic acids and melanoids, can, through the competitive interactions of hydroxyl groups with positively charged lecithin fragments, cause the weakening of the alginate–lecithin interactions forming the film structure, which leads to destabilization of polymer network of the film, resulting in weakening of its strength (which leads to a reduction in the values of the mechanical parameters of the film). In contrast, the ALM film shows better mechanical properties (MBL and TS) (Table 5) compared to other films and the control film. Therefore, medium roast coffee extract can be considered a crosslinking agent that improves the mechanical properties of the film. These unique properties of medium roast coffee extract may be due to the fact that this type of extract presumably contains comparable amounts of two groups of bioactive antioxidants, (chlorogenic acids) and melanoidins formed during the roasting of the coffee beans. In this case, it is most likely that the strength of the interactions created between the melanoids and chlorogenic acids balances the strength of the competitive interactions between these components of the coffee extract and the structure-forming film. So, it is not weakened, unlike when one group of these compounds predominates in the extract. These results are partly justified by the literature data, according to which individual antioxidants can have a positive or negative effect on film strength depending on the structure of the phenolic compound and its concentration in the film. Phenolic compounds, such as tert-butylhydroquinone or quercetin, when added to cassava starch/gelatin film at high concentrations, result in a decrease in the tensile strength parameter, which is due to the formation of hydrogen bonds between hydroxyl groups of antioxidant and gelatin/starch molecules [49]. On the contrary, an addition of tea-derived polyphenols to gelatin/sodium alginate film resulted in an increase in the value of the TS parameter, which means that tea-derived polyphenols may serve as crosslinking agents for protein/polysaccharide polymers [10]. These results confirm that extracts rich in phenolic compounds used as additives to polysaccharide films can have a positive or negative effect on the mechanical properties of the edible film, depending on their structure, and especially on their molecular weight and polarity. In our opinion, an observed decrease in TS values for the foils with the addition of coffee extracts, with the exception of one with medium roasting coffee, may result, to a great extent, from the weakening of interactions between the primary components of the film, i.e., alginate and lecithin, which was reflected in a decrease in the enthalpy of the melting ΔH_m_ for the samples with an addition of the extracts when compared to the pure alginate/lecithin film. According to Ismallayli et al. [6], the tensile strength and resistance to pH changes of alginate/chitosan film were higher than that of alginate film and chitosan film. In the case of some alginate/lecithin films (i.e., those with green, light- and dark-roasted coffee), the decrease in TS value may be caused by the weakening of the foil structure by those components of coffee extract that are not classified as phenolic compounds and were not investigated in this study. In coffee extract, besides phenolic compounds and melanoidins, the composition is dominated by disaccharides and monosaccharides [50], lipids including mainly triglycerides, sterols, fatty acids, pentacyclic diterpenes, diterpenic alcohols, triterpenic esters and ceramide [51]. Also, tocopherols together with tocotrienols are present in coffee extract [52]. Other compounds include volatile and non-volatile aliphatic acids, trigonelline and nicotinic acid. In the case of strong-roasted coffee extract, also the guaiacol derivatives as well as organic sulfur compounds resulting from the pyrolysis of the coffee beans’ constituents are present [28].

The modulus of elasticity (ME), i.e., the mechanical parameter that determines elasticity, was also specified. The value of the ME ranged from 3693.2 MPa for the film with green coffee to 6709.9 MPa for the reference film (Table 5). The coffee extract added to the alginate/lecithin film results in a decrease in ME values, but the decrease is the smallest for the film with medium roast coffee, for which the ME is equal to 5749.1 MPa.

### 2.9. Water Absorption and Solubility

The water content in films is an indicator of the film’s hydrophilicity. According to researcher [53], the difference in moisture level may have occurred due to the change in the hydrophilicity of the films. In this case, it may have an impact on the water permeability properties of the edible films. The results regarding the water properties of the alginic, alginic/lecithin and films with coffee extract of different degrees of roasting are presented in Table 5.

The water content analysis of all samples demonstrated that the values were within the range of 11.29–17.31%. The integration of coffee extracts had a significant influence on the moisture content of the films. The reduced water solubility of the ALM and ALD films is due to the lower content of free phenolic acids (FPAs), TPCs and AAs and therefore the higher content of melanoid compounds.

The solubility of the obtained films was 100% regardless of the additives. These results are in line with the data available in the literature on the subject, estimating the solubility of alginic acid as 100%.

Due to their total water solubility, alginate/lecithin films, regardless of the type of added extract, can be used as edible films where solubility is one of the most important properties in food or pharmaceutical applications.

### 2.10. Water Contact Angle Determination

The water contact angle (WCA) is considered an important indicator that is used to assess the level of hydrophilicity or hydrophobicity of the film surface. A film surface is considered hydrophilic if the WCA is low (below 90°) and hydrophobic if it is higher (above 90°). The results show that the type of extract used for the lecithin/alginate film affects the WCA value. It reaches the lowest value for the ALG film (32.46) and the highest value for the ALD film (41.30) (Table 5). The observed increase in film hydrophobicity that accompanies an increase in coffee roasting level indicates a significant difference between the physicochemical nature of polyphenols present in green coffee and the melanoids that predominate in dark roast coffee. The chlorogenic acids present in the ALG film exhibit strong polarity, which influences their ability to form hydrogen bonds between components of the alginate/lecithin film and water molecules. These results in a clear advantage of the hydrophilic properties of this film over the other films tested (Table 5). In contrast, macromolecular melanoids show a much lower tendency to form hydrogen bonds with film components and water molecules. This results in an increase in the hydrophobicity of the film.

### 2.11. Foil Barrier Properties Estimated Against UV–Vis Light

The barrier properties of edible films and packaging against UV–Vis light play a very important role. Visible and ultraviolet light is responsible for decomposition processes and especially for oxidation of food components, which results in adverse changes in the appearance, taste and smell of food products and a reduction in nutritional value. Transmission spectra in the range of light (200–800 nm) were performed for the lecithin/alginate films tested (Figure 3). The obtained spectra revealed that the addition of coffee extract resulted in a significant improvement in the barrier properties of the films against UV light, irrespective of the type of extract added. That phenomenon is reflected by a drastic decrease in the transmittance values of the tested samples in the range from 200 to 400 nm with respect to the control film (AL). In the case of the films with coffees of different roasting levels, the coffee extract was also found to affect the barrier properties of the film against visible light (400–700 nm). This effect was observed to the greatest extent for the ALM and ALD films and to the least extent for the ALG and ALL films. This indicates that melanoids play the largest role in shaping barrier properties against Vis light. This fact confirms the benefits of using the new type of natural additive for alginate–lecithin films, the purpose of which is to protect the product from the harmful effects of sunlight.

### 2.12. Color Parameters and Opacity of the Films

Based on the obtained results, it was found that adding coffee extract to alginate/lecithin film results in a decrease in lightness (L*) compared to the extract-free baseline film (AL) (Table 6), as well as an increase in the values of the color parameters determining the balance between red and green (a*) and between yellow and blue (b*). Significant differences between the values of color parameters for the films were observed depending on the type of extract used. The film with green coffee extract was the lightest (L* = 83.33), while the one enriched with dark roast coffee extract was the darkest (L* = 64.21), which was due to the highest concentration of dark melanoids in the latter sample. The films containing extracts of roasted coffee were more reddish and yellowish compared to the film with green coffee. In contrast to lightness, the highest values of a* and b* were found for the film enriched with dark roast coffee extract. These observations were confirmed for the values of the WI (whiteness index) and YI (yellowness index) (Table 6) calculated with the use of the measured L*, a* and b* values. It was found that the non-enriched control film had the highest WI (87.03), while the sample with dark roast coffee extract had the lowest WI (48.31). The opposite trend was observed for YI, with the smallest value for the control film (16.46). In order to characterize the brown color of the films with the addition of roasted coffee extract, the browning index (BI) was calculated for all the samples. The obtained results confirmed the fact that values of the BI increased along with the increase in the degree of roasting. The sample with green coffee extract was the lowest (23.08) among the films with the extract addition, whereas the film enriched with dark-roasted coffee extract was the highest in this respect (88.53).

In addition, the value of the total color difference (ΔE) was calculated (Table 6). In contrast to the previously discussed color parameters, it was found that the film with medium roast coffee had the highest value of ΔE relative to the control film (35.58), while the film with green coffee had the lowest value (11.66). Nevertheless, the calculated ΔE values indicate clear differences in color between the control film and the samples enriched with coffee extracts.

### 2.13. Determination of TPC and AA of the Coffee Extracts and Films

The content of TPC in the studied films is shown in Figure 4A. It was stated that alginate/lecithin films enriched with coffee extracts differed significantly in terms of this parameter. The values of TPC for the tested samples, expressed in GAE, ranged from 172.83 to 275.12 μg GAE per 1 g of film. The ALL film was characterized by the highest TPC (275.21), while the ALD was the lowest (146.34). Taking into account the content of antioxidants, the ALL film was characterized by the most favorable proportion between chlorogenic acids and melanoidins, both responsible for the ability to reduce the Folin–Ciocalteu reagent by components of extracts added to the films. The tested films were also characterized by a varied AA measured in reaction with DPPH radicals (Figure 4B). The ALL film had the highest AA value (1.652 µM/T), while the ALD film had the lowest (0.827 µM/T), respectively. A statistically significant correlation was found between the content of the TPC and AA of the films (r = 0.981). Based on the results obtained, it can be concluded that an addition of coffee extracts to alginate/lecithin films significantly affects the antioxidant properties of this type of edible film.

### 2.14. Cytotoxicity and Viability Analysis

A human epithelial cell line (Caco-2) derived from a colon carcinoma (ATCC^®^ HTB-37™), a human hepatocyte carcinoma cell line (HepG2) (ATCC^®^ HB-8065™) and a human foreskin fibroblast cell line (BJ) (ATCC^®^ CRL-2522™) were treated with final concentrations (0.1 μg/mL->100 μg/mL) of the digested films (AL, ALG, ALL, ALM and ALD). As indicated in Figure 5A–I, all concentrations of films were below IC_50%_ at all time points (24, 48 and 72 h), suggesting the safety of the films even at the highest concentration (100 μg/mL). In the Caco-2 cell line, the highest cytotoxicity (31.69 ± 0.05) was observed for the AL film after 24 h incubation (Figure 5A) with the highest dose of 100 μg/mL, while after 48 and 72 h (Figure 5B,C), it reached −11.50 ± 0.17 and −11.53 ± 0.32, respectively. The lowest cytotoxicity (−76.97 ± 0.02) for the Caco-2 cell line was observed for the ALD film after 24 h in a dose of 100 μg/mL.

For the HepG2 and BJ lines, the cells were most sensitive to the AL film in the 10 μg/mL dose (18.84 ± 0.63) after 72 h (Figure 5F) for the HepG2 line and 1 μg/mL (27.02 ± 0.18) after 24 h for the BJ line (Figure 5G). The lowest levels of cytotoxicity for both lines were observed for the ALM films in a concentration of 1 μg/mL after 24 h (−30.07 ± 0.12) for HepG2 cells (Figure 5D) and after 48 h incubation with 0.1 μg/mL (−70.55 ± 0.09) for the BJ cell line (Figure 5H).

In the cell viability analysis only in the Caco-2 cell line, after a 72 h treatment with 10 μg/mL of digested AL film, a statistically significant decrease in cell viability (% of control: 87.96 ± 4.31), as compared to untreated cells (negative control), was observed (Figure 6C).

In contrast, a statistically significant increase in cell viability relative to the negative control was noted for all cell lines (Figure 6A–I). The highest increase in viability was observed after 48 h in a dose of 1 μg/mL for the ALM film (% of control: 194.43 ± 38.30), after 24 h in a concentration of 10 μg/mL of ALD (% of control 191.67 ± 26.09) and after 72 h in a dose of 10 μg/mL of the ALM film (% of control 174.95 ± 3.81) for Caco-2, HepG2 and BJ cell lines, respectively.

In our earlier study, we showed that alginate-based biocomposites with the addition of antioxidant plant extracts did not exhibit cytotoxic properties (IC_50_ not reached) and had no negative effect on HepG2 and BJ cell proliferation [54].

Hutachok et al. [55] performed a study on viability using an MTT assay in eripheral blood mononuclear cells (PBMC) and human breast cancer cells (MDA-MB-231 and MCF-7). PBMC, MDA-MB-231 and MCF-7 cells treated with roasted coffee extracts demonstrated that they were neither toxic to normal mononuclear cells nor breast cancer cells. PBMC cell viability after 48 h treatment with all extracts was significantly increased in a concentration-dependent manner.

The study on the cell cytotoxicity of green coffee bean extracts (GCBEs) reported that 10 µg/mL concentration showed differences after 24 h of treatment in OE-33 and T24 cancer cells, while 100 µg/mL GCBE showed little effect, and only the highest dose (1000 µg/mL) affected the viability of normal CCD-18Co cells with reductions ranging from 87% (2 h exposition) to only 22% in the Caco-2 cancer line (24 h exposition) compared to the control cells [19].

Additionally, in genotoxicity studies using the Cytokinesis-Block Micronucleus Assay on HepG2 cells, none of the coffee extracts of different degrees of roasting from the commercial and special quality coffee evaluated were able to induce clastogenicity or aneugenicity in the tested cells. For all samples in all tested concentrations, the CBPI values ranged from 1.70 to 1.83, which indicated that most of the treated cells were binucleated. The obtained RI values for all the samples at all tested concentrations showed that most of the treated cells were normally divided. Also, the inhibition of cell growth was up to 7.2%. This indicated that none of the coffee samples showed cytotoxicity of HepG2 cells at the treated concentrations [18].

### 2.15. Influence of Digestion of Film to Adhesion of Lactic Acid Bacteria and Anti-Inflammatory Response in Murine Macrophages on Their Produce Nitric Oxide

For adhesion assays, 10% and 5% of the digest (at concentrations of 250 and 125 µg film/mL, respectively) were introduced to the cell lines along with *L. rhamnosus* and *L. gasseri* (Figure 7). Notably, for *L. rhamnosus*, all materials exhibited a reduction in adhesion ranging from 14% to 60% at a 10% concentration of digestate. At a 5% concentration of digestion, adhesion was similarly decreased by 25% to 53%, except for the ALM film, which showed a contrary increase of 36.6% in adhesion. In the case of *L. gasseri*, the reduction in adhesion was more pronounced at a 10% digestive film concentration, ranging from 35% to 60%, and at a 5% digestive film concentration, it was reduced by 32% to 61%.

As reported by Grzelczyk et al. [56], there is a 45–55% reduction in bacterial viability during digestion depending on the type of coffee. This could also affect the intrinsic adhesion of the bacterial strains we added and reduce their ability to adhere better. However, an extended digestion time might lead to a higher release of bioactive compounds from the film, potentially positively influencing the gut microbiota [56]. Therefore, digestion plays a crucial role in releasing polyphenols into the gut environment and influencing their interaction with the gut microbiota.

Polyphenols modulate the gut microbial population, and their metabolism by the microbiota alters their bioavailability. Higher molecular weight polyphenols, fully available for metabolism in the hindgut, are particularly significant for host health [57]. The interplay between dietary polyphenolic bioactives and the gut microbiota is crucial to health outcomes. The interactions involve how polyphenols are metabolized by the microbiota and how they can modulate the microbiota to enhance their impact on preventing and improving certain [58]. While the gut microbiota influences polyphenol metabolism and produces bioactive metabolites, polyphenols also shape the composition of the microbiota [59].

The subsequent effect under investigation was the impact of digestion on the ability to inhibit nitric oxide production in mouse macrophages of the RAW 264.7 cell line when stimulated with LPS. According to the findings, all the materials tested (at 20% digestion or 500 µg/mL sample concentration) exhibited a reduction in nitric oxide (NO) production. Nevertheless, compared to the internal control, only the ALL sample displayed a noteworthy effect in significantly decreasing NO production. A statistically significant reduction of more than 31% in NO production was observed with a p-value less than 0.05 [60]. As indicated by previous studies, chlorogenic acid has been observed to suppress LPS-induced NO expression in the RAW264.7 cell line, as reported by Kim et al. [60]. As reported by Funakoshi-Tago et al. [61], in a dose-dependent manner, coffee extracts significantly reduce NO production in the same cell line, even by more than 80% at a dose of 5% *v*/*v*. Similarly, the same is true for iNOS mRNA production [62,63].

### 2.16. Microbiological Tests

The results of the microbiological analyses are presented in Table 7. Each of the tested films with the addition of extract from green coffee beans or coffee beans of different degrees of roasting had growth inhibitory properties towards selected species of bacteria.

Analysis of ALG film showed growth inhibition of *E. coli*, *S. aureus*, *S. equorum*, *S. xylosus* and *E. faecalis*. Similar results were observed by other scientists for *S. aureus* and *S. enteritidis* [64]. These results are very important, mainly due to the fact that *E. coli* and *S. aureus* as potentially pathogenic bacteria, which often serve as model bacteria in studies on the bacteriostatic and bactericidal properties of various compounds [63,65,66,67].

Analysis of the ALD and ALM films showed growth inhibition of *B. megaterium*, *S. equorum*, *S. xylosus*, *E. faecalis* and *S. typhimurium* (Table 7). Analysis of ALL film showed growth inhibition of *B. megaterium*, *S. equorum*, *S. xylosus* and *E. faecalis* (Table 7). The obtained results are consistent with the results of the research conducted by Daglia et al. [68], where all the tested samples of roasted coffee (light, medium and dark) showed a clear antibacterial effect depending mainly on the roasting process.

Our studies showed that the largest zone of inhibition for each tested film with coffee extract was observed for *S. aureus*, but growth inhibition was not observed for *B. thuringiensis*, *B. cereus*, *M. luteus* and *S. enteritidis*.

Compounds contained in coffee can inhibit the growth of bacteria by changing the structure and function of the bacterial cytoplasm [64]. It was found that caffeine contained in coffee at a concentration of 62.5 to >2000 μg·mL^−1^ can inhibit the growth of bacteria, while a higher concentration (>5000 μg·mL^−1^) inhibits the growth of mold [69].

The antibacterial activity of films with coffee extract against pathogenic bacteria was confirmed by Dondapati et al. [70] and Rante et al. [71]. It was found that coffee has significant activity against the growth of food spoilage bacteria [72].

The antimicrobial activity of coffee extracts was described inter alia against *E. coli*, *S. typhi*, *P. aeruginosa*, *S. aureus*, *B. cereus* and *S. faecalis* [17,73,74,75]. However, the results obtained in our research are contradictory and indicate a lack of antimicrobial activity of the studied films against *B. cereus*. On the other hand, the research conducted by Duangjai et al. [75] showed the antibacterial activity of coffee pulp extracts (*Coffea arabica* L.) against both Gram-positive bacteria (*S. aureus* and *S. epidermidis*) and Gram-negative bacteria (*P. aeruginosa* and *E. coli*) [76]. In the case of *S. typhimurium*, zones of growth inhibition were observed only for the ALD and ALM films.

## 3. Materials and Methods

### 3.1. Chemicals

Acetic acid (99.5%, CAS No: 64-19-7) was purchased from Pure Land (Stargard, Poland), sodium hydroxide (CAS No: 1310-73-2) from Stainlab (Lublin, Poland) and ascorbic acid (99%, CAS No: 50-81-7) from Chempur (Piekary Śląskie, Poland). Hydrochloric acid (35–38%, CAS No: 7647-01-0) and ethyl acetate (98%, CAS No: 141-78-6) were purchased from Stainlab (Lublin, Poland). Sodium chloride (99.5%, CAS No: 7647-14-5) was acquired from LOBA CHEMIE PVT.Ltd. (Mumbai, India). Ethylenediaminetetraacetic acid (EDTA) (99%, CAS No: 60-00-4) was acquired from Sigma-Aldrich (Saint Louis, MO, USA). Folin–Ciocalteu’s reagent (CAS No: 7647-01-0) and anhydrous sodium carbonate (99.5%, CAS No: 497-19-8) were bought from Chempur (Piekary Śląskie, Poland). *6*-Hydroxy-2,5,7,8-tetramethylchroman-2-carboxylic acid (Trolox, 97%, CAS No: 53188-07-1) and 2,2-diphenyl-1-picrylhydrazyl (DPPH, 97%, CAS No: 1898-66-4) were purchased from Sigma-Aldrich (Saint Louis, MO, USA). Gallic acid (97.7–102.5%, CAS No: 149-91-7) was purchased from Sigma (Steinheim, Germany). Caffeic acid (98.0%, CAS No: 331-39-5), *p*-coumaric acid (98%, CAS No: 501-98-4) and ferulic acid (99%, CAS No:99537-98-4) were purchased from Sigma (Steinheim, Germany). The monocaffeoylquinic acids, including 5-O-caffeoylquinic acid (95%, CAS No: 327-97-9), 4-O-caffeoylquininc acid (98%, CAS No: 905-99-7), 3-O-caffeoylquinic acid (98%, CAS No: 906-33-22), were purchased from Sigma (Steinheim, Germany). The dicaffeoylquinic acids, including 3,5-di-caffeoylquinic acid (95%, CAS No: 2-450-53-5), 3,4-di-caffeoylquinic acid (90%, CAS No: 1-4534-61-3) and 4,5-di-caffeoylquinic acid (85%, CAS No: 5-7378-72-0), were purchased from Sigma (Steinheim, Germany). Acetonitrile of the gradient grade (HPLC) (99.9%, CAS No: 75-05-8) was purchased from Supelco (Darmstadt, Germany). Sodium alginate (Sigma Aldrich, CAS No.: 9005-38-3), glycerol (StanLab, Lublin, Poland, p.a. CAS No.: 56-81-5), lecithin (95% phosphatidylcholine, Louis Francois, Marne La Vallee, France), sodium nitrate (Eurochem, p.a. CAS No.: 7631-99-4). NaCl (StanLab, Poland, CAS: 7647-14-5), TSA LAB-AGAR (Biomaxima, Poland, REF. PS 22). Eagle’s Minimum Essential Medium (EMEM; ATCC, Manassas, VA, USA, Cat Num: 30-2003), fetal bovine serum (FBS, Cat Num: 30-2020) were obtained from American Type Culture Collection (Manassas, VA, USA), Cytotoxicity Detection Kit (LDH) (Cat Num: 11644793001, Roche, Basel, Switzerland). Fluorescein isothiocyanate (Life Technologies, Carlsbad, CA, USA; CAT. Num.: 119250010), PBS (VWR, France, Cat. Num.: 392-0434), RPMI1640 medium (BioWest, France, Cat Number L0505-500), non-essential amino acids (BioWest, France, Cat Num.: X0557-100), glucose (Sigma-Aldrich, Saint Louis, Missouri, USA, Cat. Num.: G8644-100mL), Griess reagent (Sigma-Aldrich, Saint Louis, Missouri, USA, Cat. Num.: G4410-10G), lipopolysaccharide (LPS, Sigma-Aldrich, Saint Louis, Missouri, USA, Cat. Num.: L2630-10MG).

### 3.2. Coffee Beans

The light-, medium- and dark-roasted coffee beans originating from the same batch of green beans Brasilia Santos Arabica coffee beans (Bourbon variety) were roasted in the local coffee roasting plant located in Krakow (Poland).

### 3.3. Preparation of Coffee Extract

Extraction of phenolics from the coffee samples was performed according to Andrade et al. [27] with minor modifications. The 5 g of green and roasted coffee beans were homogenized, and then the powdered samples were mixed with 60 mL of ethanol/water mixture (40/60, *v*/*v*) for 24 h. Obtained mixture was filtered and stored in the dark at 4 °C.

### 3.4. Carbohydrate Polymer Characterization

Sodium alginate (extracted from the alga *Macrocystis pyrifera*) was purchased from Sigma Aldrich (St Louis, MO, USA). The viscosity of sodium alginate was 15.0–25.0 cps., mannuronate/guluronate ratio of 1.58, obtained from FT-IR spectroscopic analysis [77].

### 3.5. Films Preparation

Soya lecithin powder, GMO-free, 95% phosphatidylcholine (Louis Francois, Marne La Vallee, France) was dissolved in distilled water and stirred at 300 rpm at room temperature overnight. The film formulations were made by dissolving sodium alginate viscosity 15–25 cps (Sigma Aldrich, St Louis, MO, USA) (2.5%, *w*/*v*) in deionized water while stirring at temperature of 70 °C, and then glycerol (StanLab, Lublin, Poland) (1%, *w*/*w*) was added. Next, dissolved lecithin (5%, *w*/*w*) was added drop-by-drop to the final solution, and mixture was stirred for 30 min at room temperature. The final solution was then homogenized with laboratory homogenizer (15,000 rpm) for 5 min. Subsequently, after drying the final solution (alginate 2.5%, *w*/*v*; lecithin 5%, w/w and glycerol 1%, *w*/*w*), the control film (AL) was obtained. For the preparation of AL–coffee extract complexes, to the portions of AL solution were added ethanol/water mixtures of the four different coffee extracts (5%, *v*/*v*). All solutions of AL–coffee extracts (green (G), light (L)-, medium (M)- or dark (D)-roasted coffee) were homogenized with a laboratory homogenizer (15,000 rpm) for 5 min. Finally, the samples were transferred into Petri dishes (diameter 14 cm, volume of solution: 50 cm^3^) and dried at the same conditions and time (48 h at room temperature under the fume cupboard) as AL. In this manner, the four complexes with different additions of coffee extracts were obtained (ALG, ALL, ALM and ALD).

### 3.6. Determination of Total Phenolic Content (TPC) in the Coffee Extracts

The TPC was determined using a Folin–Ciocalteu reagent following the procedure described by Singelton and Rossi [78]. The measurements were performed in triplicate. The TPC was calculated as mg of gallic acid equivalents per 1 dm^3^ of the coffee extract.

### 3.7. Determination of Antioxidant Activity (AA) in the Coffee Extracts

Determination of AA was performed in the reaction with DPPH radical, according to the procedure described by Blois [79]. The measurements were performed in triplicate. The AA was calculated as mM of Trolox equivalent per 1 dm^3^ of coffee extract.

### 3.8. Determination of Chlorogenic Acid (CGA) Content in the Studied Coffee Extracts

The phenolic profile of coffee extracts was performed using high-performance liquid chromatography (HPLC) according to the procedure developed by Fujioka and Shibamoto [26]. The determination of chlorogenic acid content was performed by the use of an HPLC apparatus (HPLC, Jasco, Tokyo, Japan) equipped with a DAD detector (MD-2018 plus, Jasco, Japan). The mobile phase A was citric acid, and mobile phase B was methanol. The linear gradient was initially set at A/B ratio of 85:15 from 0 to 5 min, then linearly increased to 60:40 at 40 to 85 min. The flow rate was 1.0 cm^3^/min.

The chromatographic analysis was carried out on a Spherisorb (ODS) column (250 mm × 4 mm, particle size 5 µm) at a temperature of 30 °C and a flow rate of 1 cm^3^/min. The qualitative and quantitative analyses of chlorogenic acids (i.e., chlorogenic, cryptochlorogenic and neochlorogenic acids, as well as 3,4-dicaffeoylquinic, 3,5-dicaffeoylquinic and 4,5-dicaffeoylquinic acids) were made.

### 3.9. Determination of Free Phenolic Acids (FPA) in the Studied Coffee Extracts

In order to estimate the total content of FPA by means of HPLC including free forms of phenolics (i.e., caffeic, p-coumaric and ferulic acids released from their bound forms, i.e., chlorogenic acids) in the coffee extract, a hydrolytic procedure was performed according to Nardini and Ghiselli [80]. Namely, 10 cm^3^ of the coffee extract was mixed with 90 cm^3^ 2 M NaOH water solution with an addition of 1% of ascorbic acid and 10 mM per dm^3^ of EDTA. Subsequently, the obtained mixture was incubated for 30 min at temp. of 30 °C and then neutralized using HCl solution (1:2, *v*/*v*). The neutral solution was saturated by sodium chloride and then triple extracted using ethyl acetate. After the extraction process, the organic solvent was evaporated to dryness under reduced pressure and the obtained dry residue was dissolved in methanol. The qualitative analysis of phenolic acids was carried out using a DAD detector (MD-2018 Plus, Jasco, Japan). The calibration curves of analyzed phenolic acids were made in triplicate for each individual standard and were plotted separately for each standard at concentration in the range of 0.02–0.2 mg/dm^3^. The analyses of phenolic acids in coffee extract samples were carried out three times.

### 3.10. Physicochemical Properties of the Films

#### 3.10.1. ATR-FTIR Spectrophotometry

The spectral measurements of the films were made with the ATR–FTIR spectrophotometer Nicolet iS5 (Thermo Scientific, Waltham, MA, USA). A MIRacle ATR accessory equipped with a ZnSe crystal was used for sampling. The FTIR spectra were recorded in the range of 4000–700 cm^−1^ at a resolution of 4 cm^−1^. All the spectra were performed at room temperature (23 ± 0.5 °C).

#### 3.10.2. High-Performance Size Exclusion Chromatography (HPSEC–MALLS–RI)

The HPSEC system and methods of measurement are described in our previous publications [41,81]. Astra 4.70 software (Wyatt Technology, Santa Barbara, CA, USA) with a Berry plot third-order polynomial fit was applied for the calculation of M_w_ and R_g_ values of alginate, alginate/lecithin and another film (ALG, ALL, ALM and ALD films) [82,83].

#### 3.10.3. Differential Scanning Calorimetry (DSC) Analysis

The DSC analyses were obtained using a differential scanning calorimeter (Netzsch, Selb, Germany, Phoenix DSC 201 F1). The investigated film samples (approx. 2 mg) were closed hermetically in a standard aluminum pan and heated from 25 °C to 300 °C at a heating rate of 10 °C/min under constant purging of nitrogen at 20 cm^3^/min. An empty aluminum pan was used as the reference probe. The temperatures and enthalpy of thermal transitions were determined with the use of instrument’s software Proteus Analysis (software v. 4.8.2 (Netzch, Selb, Germany). The characteristic peak temperature and enthalpy values of endotherm and exotherm were recorded. The analyses were carried out in three replications.

### 3.11. Determination of Mechanical Properties of the Films

In order to determine the mechanical properties of the films, the following parameters were tested: maximum breaking load (MBL), modulus of elasticity (ME) and tensile strength (TS). These parameters were evaluated using an EZ-SX texturometer (Shimadzu, Kyoto, Japan) in a stretch mode (with head movement speed of 1 mm/s) and according to the ASTM D882-18 [84]. The analyses were carried out in four replications.

### 3.12. Water Content and Solubility of the Films

The water content (WC) of prepared films was measured according to the method by Souza et al. [85] with slight modifications. The films were cut into squares (3 cm × 3 cm) and weighed to the nearest ∼0.0001 g (W_1_—initial weight). Then, the films were dried in an oven at 70 °C for 24 h to obtain the initial dry matter (W_2_). After drying, samples were placed into 30 cm^3^ of Milli-Q water for 24 h. Since our films dissolved completely, no further measurements were performed. Water content was calculated with the following equation:water content [%] = (W_1_ − W_2_/W_1_) ∗ 100

### 3.13. Water Contact Angle Determination

Water contact angle (WCA) was determined according to Jamróz et al. [86].

### 3.14. Determination of Foil Barrier Properties Against UV–Vis Light

In order to evaluate the foil barrier properties, the transmission spectra of the studied films were made in the range of 200–800 nm using a V-630 spectrophotometer (Jasco, Japan).

### 3.15. Determination of the Color Parameters and Opacity

Color parameters of the films were established in the CIELAB system by the reflection method (illuminant D65, range 400–700, measuring gap 25 nm, observer 10°) using a Color i5, X-Rite spectrophotometer (X-Rite Inc., Grand Rapids, MI, USA). The measurements were carried out in 5 replicates and were presented as L* (lightness), a* (red–green balance) and b* (yellow–blue balance). The measurements were carried out on a white background using a white master plate. The total value of color difference (ΔE) was calculated on the basis of Equation (1). ΔE was calculated in relation to the control sample (lecithin/alginate film). Additionally, the following parameters, including WI (whiteness index) and YI (yellowness index), were calculated on the basis of Equations (2) and (3). In order to characterize the brown color of the films, the browning index (BI) was also calculated on the basis of Equations (4) and (5).
(1)$$\Delta E=\sqrt{{\left(L-L^{*}\right)}^{2}+{\left(a-a^{*}\right)}^{2}+{\left(b-b^{*}\right)}^{2}}$$
(2)$$WI=100-\sqrt{{\left(100-L^{*}\right)}^{2}+{\left(a^{*}\right)}^{2}+{\left(b^{*}\right)}^{2}}$$
(3)$$YI=\frac{147.86\times b^{*}}{L^{*}}$$
(4)$$BI=100\times \frac{X-0.31}{0.172}$$
(5)$$X=\frac{a^{*}+1.75L^{*}}{5.645L^{*}+a^{*}-3.012b^{*}}$$

Opacity (OP) of the films was determined using a spectrophotometr Jasco V-630 (Jasco, Tokyo, Japan) at the wavelength of 600 nm and was calculated as absorbance at 600 nm/thickness of the film (mm).

### 3.16. Determination of Total Phenolic Content

The TPC of studied films was determined using the Folin–Ciocalteu method. The small piece of the foil weighing appr. 200 mg was cut into small pieces that were placed in a test tube with the addition of 5 mL of 80% ethanol and left to shake for 24 h. The obtained mixture was filtered, and then 0.5 cm^3^ of the supernatant was analyzed according to the method developed by Singleton and Rossi [78]. The measurements were performed in triplicate. The TPC of studied films was calculated as µg of GAE per 1 g of the film.

### 3.17. Determination of Antioxidant Activity of the Films

Determination of the AA of studied films was performed using DPPH assay according to Chavoshizadeh et al. [87] after minor modifications. The small piece of the foil weighing appr. 200 mg was cut into small pieces that were placed in a test tube with the addition of 5 cm^3^ of 80% ethanol and left to shake for 24 h. The obtained mixture was filtered, and then 0.3 cm^3^ of the supernatant was mixed with 3.7 cm^3^ of DPPH methanolic solution with an absorbance value of 0.5. After 30 min. of incubation, the absorbance was measured using UV–Vis V-630 spectrophotometer (Jasco, Japan) at λ = 515 nm against methanol. The measurements were performed in triplicate. The AA of studied films was calculated as µg of Trolox equivalents (TE) per 1 g of the film.

### 3.18. In Vitro Digestion of Films

The in vitro static digestion model, conducted according to Brodkorb et al. [88], was divided into three phases (oral, gastric and intestinal). The temperature of 37 °C was carefully maintained throughout the experiment in the incubator. For the digestion, 1 g of sample was used. The oral phase lasted for 2 min, followed by the gastric and intestinal phases, both lasting 2 h each. Samples were continuously mixed using special rotational shakers. During the gastric phase, the pH was lowered to 3 using hydrochloric acid, and during the intestinal phase, it was adjusted to pH 7 using sodium hydroxide. Digestion was subsequently halted by freezing the samples at −80 °C.

### 3.19. Cell Culture

Human epithelial cell line Caco-2 derived from a colon carcinoma (ATCC^®^ HTB-37™), human hepatocyte carcinoma cell line HepG2 (ATCC^®^ HB-8065™) and human foreskin fibroblast cell line BJ (ATCC^®^ CRL-2522™) obtained from ATCC (American Type Culture Collection, Manassas, VA, USA) were routinely cultured in EMEM medium (ATCC^®^) supplemented with 10% fetal bovine serum (ATCC^®^) and antibiotic mixture.

Cell cultures were stored in standard conditions (37 °C, 5% CO_2_, 98% humidity) in an incubator (NuAire, Plymouth, MN, USA). For the differentiation experiments, Caco-2 cells were seeded (50,000 cells) into upper compartments of 12-well 0.4 μm PET Transwell inserts (Greiner), and both upper and lower compartments were filled with the cell culture medium, which was replaced every 2–3 days. The differentiation progress was assessed after the monolayers reached confluence by measuring transepithelial electric resistance (TEER) using EVOM electrode (World Precision Instruments, Sarasota, FL, USA) at the designated time points. HepG2 and BJ cells were cultured in the same conditions as Caco-2 cells.

### 3.20. Caco-2 Cells Permeability Assay

Caco-2 monolayers with an integrity equivalent to a trans-epithelial electrical resistance (TEER) higher than 200 Ω/cm^2^ were used for permeability experiments. The digested sample was placed on the top of the Caco-2 cells monolayer for 2 h. Collected filtrate was then used for cytotoxicity studies.

### 3.21. In Vitro Cytotoxicity Analysis

In vitro cytotoxicity testing was performed using Cytotoxicity Detection Kit (LDH) (Roche, Basel, Switzerland). Caco-2, HepG2 and BJ cells were seeded into 96-well plates (8000 cells/well) and left for 24 h to attach. Next, the different concentrations of digested films were added, and NADH oxidation after 24, 48, and 72 h of treatment was specified by measuring absorbance with a Multiskan Go microplate reader (Thermo Scientific, Waltham, MA USA) at 490 nm. The cytotoxicity % was calculated according to the protocol provided by the manufacturer.

Cell viability was assessed by crystal violet assay. Caco-2, Hep G2 and BJ cells were seeded into 96-well plates (8000 cells/well) and cultured for 24 h. Then, cells were incubated with different concentrations of digested films for 24, 48 and 72 h. After that, crystal violet assay was performed as described by Sularz et al. [89]. The absorbance of the sample was measured using Thermo Scientific™ Multiskan™ GO Microplate Spectrophotometer (Waltham, MA, USA) at 540 nm. The results are expressed as % of negative control (100%).

### 3.22. Effect of Digestate on the Adhesion of Lactic Acid Bacteria

The digest samples underwent a threefold examination to assess their impact on the adhesion of lactic acid bacteria to intestinal cell lines. The bacterial adhesion procedures outlined in a prior study [90] were employed with some adjustments. Human intestinal epithelial cell lines Caco-2 (ACC HBT-37) and HT29 (ACCT HBT-38) were procured from ATCC (Manassas, VA, USA). A mixed cell culture, consisting of a 9:1 ratio of Caco-2 to HT29 cells at a concentration of 2 × 10^5^, was cultivated in a 96-well plate for 14 days with periodic media replacement.

On the testing day, strains of *Lacticaseibacillus rhamnosus* (DSM20711) and *Lactobacillus gasseri* (DSM20243T) were prepared at a concentration of 10^7^ CFU/mL. The bacteria were stained with fluorescein isothiocyanate (Life Technologies, Carlsbad, CA, USA) at a final concentration of 25 µg/mL, with the staining process lasting 30 min at 37 °C in a dark environment. Following staining, the bacteria underwent three washes with PBS and were introduced into the wells at a final concentration of 10^6^ CFU/mL, along with samples containing 10% and 5% digestate. After a 2 h incubation at 37 °C in a CO_2_ incubator, the plates underwent three washes with PBS, and fluorescence measurements were taken using a Tecan Infinite M200 instrument (Tecan Group, Männedorf, Switzerland) at an excitation wavelength of 495 nm and an emission wavelength of 519 nm. The percentage (%) of adhered bacteria was determined using the following formula:Adhesion ratio (%) = 100 × A/A_0_

A is the fluorescence intensity after adherence of *L. rhamnosus* or *L. gasseri* to the cell culture; A_0_ is the initial fluorescence value measured after removal of the redundant marker.

### 3.23. Effect of Digested on Inhibition of NO

The murine macrophage cell line RAW264.7 (ATCC TIB-71) obtained from ATCC was cultured in RPMI1640 medium (BioWest, France), supplemented with 10% FBS, 1% non-essential amino acids (BioWest) and 5% glucose ((Sigma-Aldrich, Saint Louis, MO, USA, Cat. Num.: G4410-10G)). The cells were incubated at 37 °C in a humidified atmosphere of 5% CO_2_.

Nitric oxide (NO) secretion was assessed by measuring nitrite levels using Griess reagent (Sigma-Aldrich, Saint Louis, MO, USA, Cat. Num.: G4410-10G). RAW264.7 cells line was seeded at 10^5^ cells/well density into a 96-well plate. After a 2 h incubation, the cells were treated for 24 h with or without 1 μg/mL lipopolysaccharide (LPS, Sigma-Aldrich, Saint Louis, MO, USA, Cat. Num.: L2630-10MG ) and samples at a concentration of 20% digestion. The supernatant media were collected, and 50 μL of the supernatant was mixed with 50 μL of Griess reagent in a separate well of the 96-well plate. The levels of NO in the supernatant were measured by assessing the absorbance of the mixture at 540 nm utilizing a TECAN Infinite M200 plate reader (Tecan Trading AG, Männedorf, Switzerland). The obtained absorbance values were compared against a standard curve generated with NaNO_3_, as described in a prior study by Paudel et al. [91].

### 3.24. Microbiological Tests

Microbiological analyses were performed based on the disc diffusion method. For microbiological tests, 9 strains of environmental bacteria were used: 3 strains of Bacillus genus (*B. thuringiensis, B. cereus, B. megaterium*), 3 strains of Staphylococcus genus (*S. aureus*, *S. equroum*, *S. xylosus*), 1 strain of Escherichia genus (*E. coli*), 1 strain of Enterococcus genus (*E. faecalis*), 1 strain of Micrococcus genus (*M. luteus*) and 2 strains of Salmonella genus (*S. typhimurium*—ATCC 14028 and *S. enteritidis*—ATCC 13076, Biomaxima, Poland). Identification of species of environmental strains was carried out by using a modern ionization method of the sample combined with measurement of its mass—MALDI-Tof-MS system, using the Bruker Daltonic MALDI Biotyper. Then, 18 h bacterial cultures were used for preparation of the suspensions of microorganisms in NaCl (0.85%) with an optical density of 0.5 McF. Next, the Petri plates were inoculated (with the tryptic soy agar medium, TSA, Biomaxima, Poland) with prepared suspensions and after drying discs were applied (6 mm diameter) with 25 µL of tested samples (25 µL of sterile deionized water for control samples). The TSA plates were incubated for 18 h at 37 °C (for *E. coli* 18 h at 44 °C). After incubation, the diameter (mm) of zone of growth inhibition of the microorganisms (including the diameter of the disc) was measured.

### 3.25. Statistical Analysis

The results are expressed as mean values ± standard deviations. The data were subjected to an analysis of variance (ANOVA), followed by Tukey’s post hoc test, *t*-test or Fisher test, which were performed to determine whether differences were significant at *p* < 0.05. Additionally, Pearson’s correlation was performed for the merged data from Table 1 and Table 2. Statistical analyses were performed with the Statistica 13.3 PL program (StatSoft, Inc., Tulsa, OK, USA) and GraphPad prism v10 (GraphPad Software, Boston, MA, USA).

## 4. Conclusions

In conclusion, new types of alginate–lecithin films were obtained with the addition of antioxidant-rich extracts from green coffee beans and coffee beans with different degrees of roasting.

The results from the ATR-FTIR and DSC analyses confirmed the interactions, both between the functional groups of the lecithin- and alginate-forming films and the interactions between the phenolic groups of the chlorogenic acids derived from the coffee beans with the mentioned components of the tested films. Films with the addition of coffee extracts gained important antioxidant properties.

Furthermore, all films, regardless of the type of additive used, gained beneficial barrier properties against UV radiation. This makes them suitable for use as packaging to protect food products, both from light radiation and the oxidative processes occurring on the surface of these products under the influence of oxygen and other negative physicochemical factors.

Cell line studies have shown that digested films do not negatively affect the viability of the Caco-2, HepG2 and BJ cells, and they do not have cytotoxic properties. This may indicate their safety as edible food packaging. Nevertheless, further studies on different cell lines and animal models are needed.

However, they negatively impact the colonization ability of selected lactic acid bacteria in the digestive tract while positively influencing the anti-inflammatory response. Specifically, they demonstrate significant capability in reducing nitric oxide production in the RAW264.7 cell line.

Based on the obtained results, the tested films with coffee extracts showed selected antibacterial activities, which makes coffee a promising potential natural food ingredient that extends the shelf life of food products. This biopolymer–phospholipid combination with the addition of ethanol–water extracts of coffee polyphenols appears to be promising and requires further research.

In our opinion, the foil characterized by the best properties is the one enriched with light-roasted coffee extract. Firstly, this foil is characterized by the highest TPC and AA contents, which, especially in the case of the last parameter, plays an important role in preventing food from oxidation and extending its shelf life. Secondly, among all the enriched alginate/lecithin foils under study, the foil enriched with light coffee extract is characterized by the most balanced values of some color parameters, including whiteness, yellowness and browning indices as well as lightness.

Alginate–lecithin films with coffee bean extracts are environmentally friendly packaging. Likewise, their production does not require the use of harmful and environmentally hazardous chemical reagents.

## Figures and Tables

**Figure 1 ijms-25-12093-f001:**
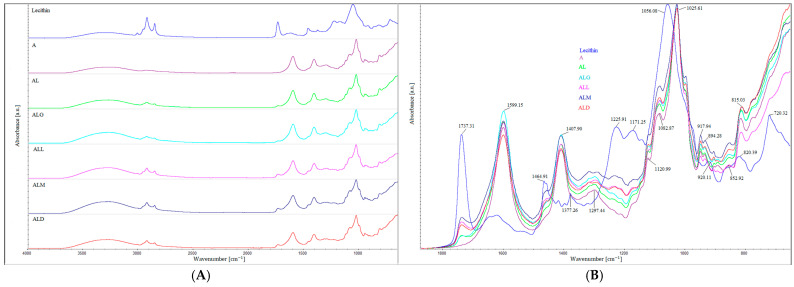
The FTIR spectra of the films (**A**,**B**) with coffee extract of different degrees of roasting (A—alginate, AL—base foil, ALG—green coffee, ALL—light roasted, ALM—medium roasted, ALD—dark roasted).

**Figure 2 ijms-25-12093-f002:**
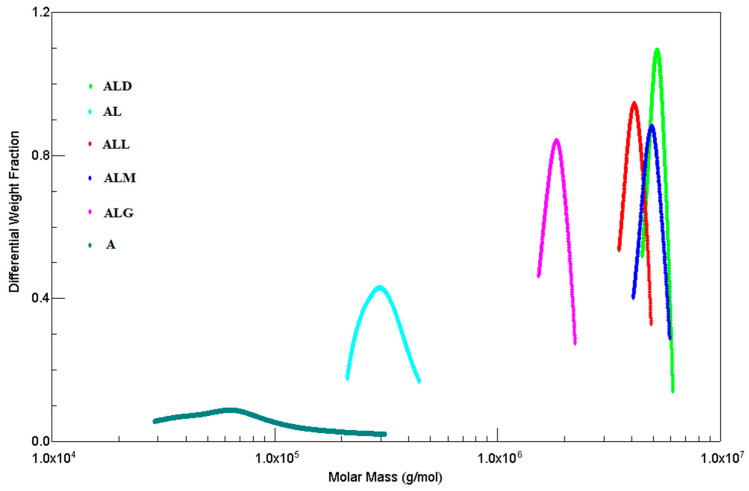
Plots of differential weight fraction vs. elution volume for the alginate, alginate/lecithin and alginate/lecithin films with coffee extracts of different degrees of roasting.

**Figure 3 ijms-25-12093-f003:**
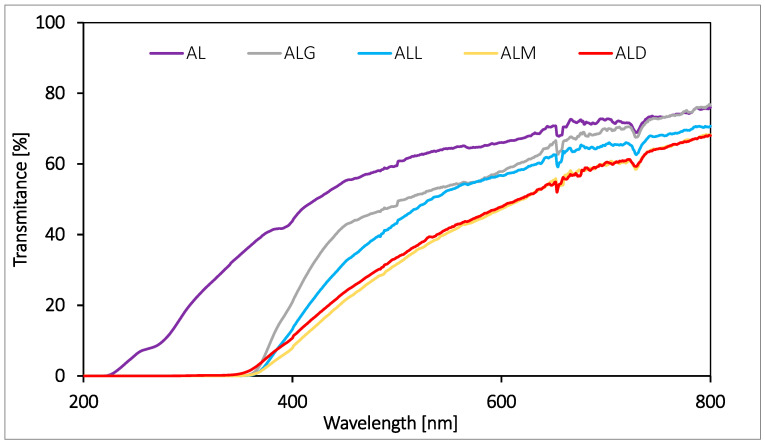
The transmission spectra of studied films including reference film (AL) and the films with the addition of green coffee (ALG), light- (ALL), medium- (ALM) and dark-roasted (ALD) coffee extracts.

**Figure 4 ijms-25-12093-f004:**
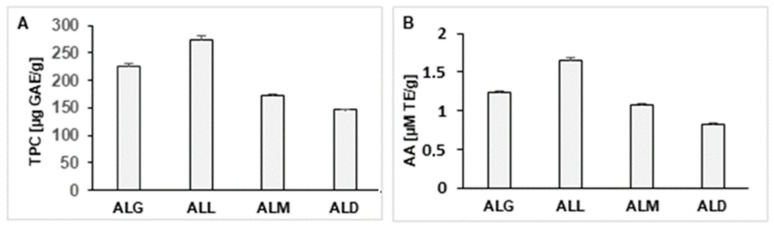
Total phenolic content [µg/g] (**A**) and antioxidant activity [µM/g] (**B**).

**Figure 5 ijms-25-12093-f005:**
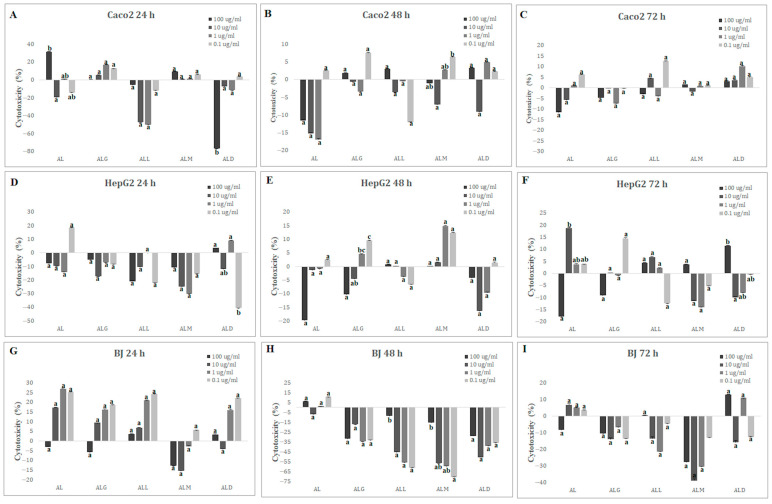
Effect of 24, 48 and 72 h exposition to different concentrations (100–0.1 μg/mL) of alginate–lecithin films (AL) and films with the addition of green, light-, medium- and dark-roasted coffee bean extracts (ALG, ALL, ALM and ALD) on membrane permeability (LDH assay) in Caco-2 (**A**–**C**), HepG2 (**D**–**F**) and BJ (**G**–**I**) cell lines. Data are presented as means ± SD. Statistically significant differences were investigated using one-way ANOVA followed by Tukey’s post hoc. Columns with the same letters are not significantly different (*p* > 0.05).

**Figure 6 ijms-25-12093-f006:**
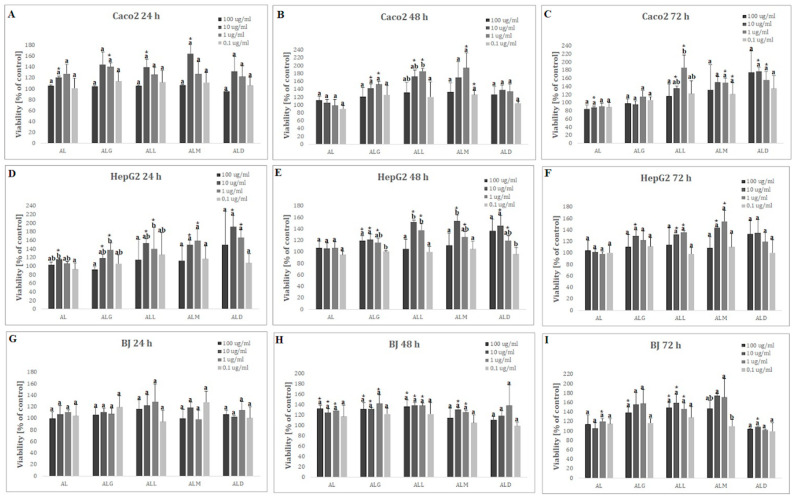
Effect of 24, 48 and 72 h exposition to different concentrations (100–0.1 μg/mL) of alginate–lecithin films (AL) and films with the addition of green, light-, medium-, dark-roasted coffee extracts (ALG, ALL, ALM and ALD) on cell viability in Caco-2 (**A**–**C**), HepG2 (**D**–**F**) and BJ (**G**–**I**). Data are presented as means ± SD. Statistically significant differences were investigated using one-way ANOVA followed by Tukey’s post hoc. Columns with the same letters are not significantly different (*p* > 0.05). * Differences statistically significant relative to the control sample when *p* < 0.05 (*t*-test).

**Figure 7 ijms-25-12093-f007:**
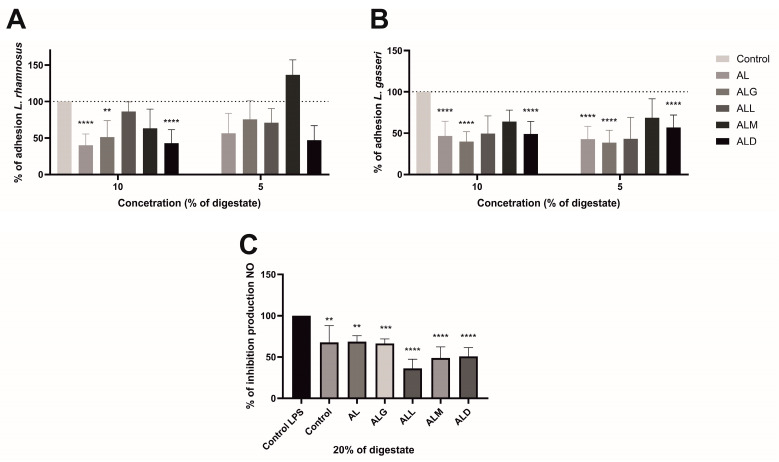
Adhesion of two lactobacilli to the mixed co-culture of Caco-2/ HT29 cell lines in the presence of two concentrations of digestion of films (**A**,**B**). Values are expressed as the mean ± standard deviation of three independent repetitions. Statistical significance is relative to the LPS-treated control (** *p* < 0.05; **** < 0.001). Film’s ability to reduce nitric production on mouse macrophages of the RAW264.7 cell line stimulated with LPS (1 μg/mL) (**C**). Values are expressed as percentage of bacterial adherence after a 2 h exposition of two concentrations of films compared to the control. Values are expressed as means ± standard error of three independent assays (** *p* < 0.05; *** < 0.01; **** < 0.001).

**Table 1 ijms-25-12093-t001:** Total phenolic content (TPC), chlorogenic acid (CGA) content and antioxidant activity (AA) of coffee extracts. ^1^ TPC values are expressed as mg of gallic acid per 1 dm^3^; ^2^ AA values are expressed as mM of Trolox per 1 dm^3^; ^3–8^ CGAs contents are expressed in mg/dm^3^ of extract. Values in the rows denoted with the same letter do not differ at the level of confidence (*p* < 0.05) and are expressed as the mean values ± standard deviations. The abbreviations of CGAs are explained in Section 2.3.

Parameter	Green Coffee	Light Roasted	Medium Roasted	Dark Roasted
TPC ^1^	2267.2 ^b^ ± 19.8	2697.20 ^d^ ± 40.6	2516.6 ^c^ ± 33.0	2057.8 ^a^ ± 20.2
AA ^2^	14.6 ^b^ ± 0.2	17.6 ^d^ ± 0.6	16.4 ^c^ ± 0.4	12.8 ^a^ ± 0.2
Chlorogenic acids				
5-CQA ^3^	2052.66 ^d^ ± 45.73	1350.17 ^c^ ± 21.92	1005.26 ^b^ ± 5.38	232.04 ^a^ ± 0.000
4-CQA ^4^	340.42 ^d^ ± 6.72	303.83 ^c^ ± 10.92	231.16 ^b^ ± 2.48	94.94 ^a^ ± 1.44
3-CQA ^5^	272.10 ^d^ ± 6.38	263.29 ^c^ ± 0.11	215.97 ^b^ ± 2.54	61.67 ^a^ ± 1.28
3,5-diCQA ^6^	343.63 ^d^ ± 4.55	69.93 ^c^ ± 0.63	45.05 ^b^ ± 1.62	6.45 ^a^ ± 0.27
3,4-diCQA ^7^	113.32 ^d^ ± 0.39	60.29 ^c^ ± 0.13	38.34 ^b^ ± 0.21	3.72 ^a^ ± 0.32
4.5-diCQA ^8^	86.31 ^d^ ± 1.70	53.46 ^c^ ± 0.36	35.44 ^b^ ± 0.22	6.24 ^a^ ± 0.21
Total CGAs	3208.44	2100.97	1571.22	405.06

**Table 2 ijms-25-12093-t002:** Free phenolic acid (FPA) content. FPA contents are expressed as mg per 1 dm^3^ of extract. Values in the rows denoted with the same letter do not differ at the level of confidence (*p* < 0.05) and are expressed as the mean values ± standard deviations.

FPAs	Green Coffee	Light Roasted	Medium Roasted	Dark Roasted
Caffeic acid	1594.74 ^d^ ± 13.39	1251.51 ^c^ ± 5.87	957.79 ^b^ ± 18.37	294.20 ^a^ ± 0.53
*p*-Coumaric acid	25.33 ^c^ ± 0.85	17.63 ^b^ ± 0.48	15.86 ^b^ ± 0.42	9.19 ^a^ ± 0.40
Ferulic acid	147.75 ^d^ ± 1.69	120.00 ^c^ ± 0.42	98.62 ^b^ ± 0.33	45.97 ^a^ ± 0.25

**Table 3 ijms-25-12093-t003:** Matrix of linear correlation coefficients for the determined values of total phenolic, phenolic acid content and antioxidant activity.

	TPC	AA	5-CQA	4-CQA	3-CQA	3,5-CQA	3,4-CQA	4,5-CQA	Total CGAs	CA **	*p*-COA **	FA **
TPC	1	0.998 *	0.373	0.597	0.715	−0.114	0.242	0.326	0.362	0.504	0.290	0.504
AA	0.998 *	1	0.441	0.628	0.745	−0.072	0.281	0.364	0.400	0.539	0.332	0.536
5-CQA	0.373	0.411	1	0.966 *	0.913	0.879	0.990 *	0.998 *	0.999 *	0.989 *	0.994 *	0.989 *
4-CQA	0.597	0.628	0.966 *	1	0.982 *	0.728	0.923	0.953 *	0.963 *	0.993 *	0.936	0.992 *
3-CQA	0.715	0.745	0.913	0.983 *	1	0.611	0.846	0.889	0.908	0.962 *	0.877	0.962 *
3,5-diCQA	−0.113	−0.072	0.874	0.728	0.611	1	0.936	0.901	0.881	0.801	0.916	0.803
3,4-diCQA	0.242	0.281	0.990 *	0.923	0.846	0.935	1	0.996 *	0.991 *	0.959 *	0.994 *	0.960 *
4,5-diCQA	0.326	0.364	0.998 *	0.953 *	0.889	0.901	0.996 *	1	0.998 *	0.980 *	0.994 *	0.980 *
Total CGAs	0.362	0.400	0.999 *	0.963 *	0.903	0.885	0.991 *	0.998 *	1	0.988 *	0.995 *	0.988 *
CA	0.504	0.539	0.989 *	0.993 *	0.962 *	0.801	0.959 *	0.980 *	0.988 *	1	0.971 *	0.999 *
*p*-COA	0.290	0.332	0.994 *	0.936	0.877	0.916	0.994 *	0.994 *	0.995 *	0.971 *	1	0.973 *
FA	0.504	0.536	0.990 *	0.992 *	0.962 *	0.803	0.960 *	0.980 *	0.988 *	0.999 *	0.973 *	1

** The contents of caffeic (CA), *p*-coumaric (*p*-COA) and ferulic acid (FA) were determined after alkaline hydrolysis, and the values denoted with * are statistically significant at α = 0.05.

**Table 4 ijms-25-12093-t004:** Molecular mass (M_w_) [g/mole] and radius of gyration (R_g_) [nm] for the alginate (A), alginate/lecithin (AL) and alginate/lecithin films with coffee extracts of different degrees of roasting (ALG—green coffee, ALL—light roasted, ALM—medium roasted, ALD—dark roasted). Values are the means of three independent experiments ± standard deviation. Parameters in columns denoted with the same letters do not differ statistically at the level of confidence (*p* < 0.05).

Sample	M_w_ × 106 [g/mol]		R_g_ [nm]	
	Peak I	Peak II	Peak I	Peak II
A	0.112 ± 0.15 ^a^	0.077 ± 0.15	95.3 ± 1.4 ^a^	59.8 ± 1.1
AL	0.308 ± 0.16 ^b^	n.m.	46.9 ± 1.4 ^b^	n.m.
ALD	5.503 ± 0.14 ^c^	n.m.	37.3 ± 1.3 ^c^	n.m.
ALM	5.184 ± 0.16 ^d^	n.m.	35.5 ± 1.4 ^c,d^	n.m.
ALL	4.227 ± 0.17 ^e^	n.m.	33.2 ± 1.4 ^d^	n.m.
ALG	1.903 ± 0.11 ^f^	n.m.	34.9 ± 1.4 ^c,d^	n.m.

n.m.—not measured.

**Table 5 ijms-25-12093-t005:** DCS measurements for pure alginate (A), alginate with lecithin (AL), alginate with lecithin and green, light-, medium-, dark-roasted coffee (ALG, ALL, ALM and ALD). The mechanical properties include tensile strength (TS), maximum breaking load (MBL), modulus of elasticity (ME), water contact angle (WCA) and water content for AL, ALG, ALL, ALM and ALD films. Parameters in columns denoted with the same letters do not differ significantly at the level (*p* < 0.05).

Sample	A	AL	ALG	ALL	ALM	ALD
Endothermic fusion (melting point)						
T_m_ (°C)	186.6 ^c^ ± 2.9	166.9 ^a^ ± 2.1	176.7 ^b^ ± 1.8	179.5 ^b^ ± 3.2	183.1 ^b,c^ ± 4.4	180.1 ^b,c^ ± 6.0
ΔH_m_ (J g^−1^)	76.96 ^a^ ± 2.49	174.53 ^c^ ± 4.77	89.86 ^b^ ± 0.87	95.85 ^b^ ± 2.01	93.43 ^b^ ± 6.85	81.45 ^a^ ± 6.02
Exothermic fusion (decarboxylation)						
T_D_ (°C)	232.4 ^a^ ± 1.5	233.9 ^a^ ± 0.8	233.3 ^a^ ± 1.0	233.8 ^a^ ± 1.7	232.5 ^a^ ± 1.0	232.5 ^a^ ± 0.8
ΔH_D_ (J g^−1^)	170.6 ^b^ ± 9.46	350.63 ^d^ ± 6.31	235.53 ^d^ ± 3.45	144.1 ^a^ ± 19.87	224.10 ^c^ ± 11.69	145.10 ^a^ ± 11.48
Mechanical properties						
TS [MPa]	-	168.9 ^b^ ± 9.0	108.5 ^a^ ± 5.9	112.9 ^a^ ± 9.8	172.3 ^b^ ± 13.6	122.9 ^a^ ± 13.0
MBL [N]	-	19.08 ^c^ ± 1.01	15.40 ^b^ ± 0.84	13.78 ^a^ ± 1.33	21.21 ^d^ ± 0.73	16.47 ^b^ ± 1.74
ME [MPa]	-	6709.9 ^c^ ± 358.8	3693.2 ^a^ ± 403.8	5357.7 ^b^ ± 190.4	5749.1 ^b^ ± 210.3	5698.0 ^b^ ± 477.7
WCA [°]	-	30.20 ^a^ ± 2.62	32.46 ^b^ ± 2.62	36.73 ^c^ ± 1.17	38.78 ^d^ ± 2.03	41.30 ^e^ ± 1.84
Water content [%]	-	17.31 ^c^ ± 0.51	13.81 ^b^ ±0.69	16.06 ^c^ ± 0.56	11.29 ^a^ ± 0.27	13.31 ^a^ ± 0.68

T_m_—melting temperature, T_D_—degradation temperature, ΔH_m_—enthalpy of melting, ΔH_D_—enthalpy of degradation.

**Table 6 ijms-25-12093-t006:** Color parameters include L*, a*, b*, WI, YI, BI, ΔE, opacity (OP) for alginate with lecithin (AL) and alginate with lecithin and green, light-, medium-, dark-roasted coffee (ALG, ALL, ALM and ALD). Parameters in columns denoted with the same letters do not differ significantly at the level (*p* < 0.05). The ΔE values are calculated in relation to the control film (AL).

Sample	AL	ALG	ALL	ALM	ALD
L*	92.69 ^e^ ± 1.06	83.88 ^d^ ± 0.73	77.12 ^c^ ± 0.72	69.38 ^b^ ± 0.89	64.21 ^a^ ± 0.67
a*	−0.86 ^a^ ± 0.03	−0.20 ^b^ ± 0.07	3.39 ^c^ ± 0.31	7.17 ^d^ ± 0.48	8.74 ^e^ ± 0.46
b*	10.68 ^a^ ± 0.32	18.26 ^b^ ± 0.52	28.56 ^c^ ± 2.02	36.34 ^d^ ± 0.78	36.26 ^d^ ± 0.38
WI	87.03	75.63	63.24	51.95	48.31
YI	16.46	31.13	52.92	74.82	80.67
BI	11.13	23.08	43.92	80.90	88.53
ΔE	-	11.66	24.09	35.58	28.11
OP	15.66 ^a^ ± 0.04	17.47 ^b^ ± 0.07	19.38 ^c^ ± 0.05	24.9 ^d^ ± 0.32	24.23 ^d^ ± 0.04

L*—lightness, a*—balance between red and green color, b*—balance between yellow and blue color, WI—whiteness index, YI—yellowness index, BI—browning index, ΔE—total value of color difference, OP—opacity.

**Table 7 ijms-25-12093-t007:** The effect of tested compounds—films with coffee extracts—on the zone of growth inhibition of microorganisms (AL—base foil, ALG—green coffee, ALL—light roasted, ALM—medium roasted, ALD—dark roasted). Averages marked with the same letters do not differ at the level of confidence by Tukey’s test (*p* < 0.05).

Microorganism	Zone of Growth Inhibition of Tested Microorganisms [mm]				
	AL	ALG	ALL	ALM	ALD
*B. thuringiensis*	0 ^a^	0 ^a^	0 ^a^	0 ^a^	0 ^a^
*B. cereus*	0 ^a^	0 ^a^	0 ^a^	0 ^a^	0 ^a^
*B. megaterium*	0 ^a^	0 ^a^	7 ^b^	7 ^b^	7 ^b^
*S. aureus*	0 ^a^	22 ^b^	0 ^a^	0 ^a^	0 ^a^
*S. equorum*	0 ^a^	16 ^b^	14 ^b^	13 ^b^	14 ^b^
*S. xylosus*	0 ^a^	8 ^b^	11 ^b^	10 ^b^	11 ^b^
*M. luteus*	0 ^a^	0 ^a^	0 ^a^	0 ^a^	0 ^a^
*E. faecalis*	0 ^a^	8 ^b^	8 ^b^	8 ^b^	10 ^b^
*E. coli*	0 ^a^	10 ^b^	0 ^a^	0 ^a^	0 ^a^
*S. typhimurium*	0 ^a^	0 ^a^	0 ^a^	7 ^b^	10 ^b^
*S. enteritidis*	0 ^a^	0 ^a^	0 ^a^	0 ^a^	0 ^a^

## Data Availability

All data are contained within this article and are available upon request.

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
