# Peer review of "Edible Alginate–Lecithin Films Enriched with Different Coffee Bean Extracts: Formulation, Non-Cytotoxic, Anti-Inflammatory and Antimicrobial Properties"

_ijms, 2024, doi:10.3390/ijms252212093_

Round 1

Reviewer 1 Report

Comments and Suggestions for Authors

I read carefully the paper “Edible alginate- lecithin films enriched with different coffee bean extracts: Formulation, non-cytotoxic, anti-inflammatory and anti-microbial properties and I found it very interesting and complete.

The point of view of the study is correct, even if it can be improved, as suggested below and also in the note reported in the attached file.

From paragraph 3.17 to 3.22 I’m not enough qualified to say if they are reported or done correctly.

Some analytical methods are well reported, while others are missing (highlighted in the attached file).

Tables and figures don’t explain themselves. Legends should be provided for each abbreviation.

Table 2. Free phenolic acids (FPAs) content. FPAs contents are expresses in mg per 1 dm3 of extract differently from what’s stated in line 716.

All the chemical compound should be named at their first appearance. This is done sometimes, not for all compounds.

The legend inside FIG.2 is not legible.

Table 3 is not well explained neither discussed…

Table 4 should be improved with a legend on the bottom.

Figure 4 add the SI units in the caption

Line 556: PBMC, MDA-MB-231 and MCF-7 if you are not so confident with these assays you can’t understand properly. So simply adding their extended name or group of application could help the reader.

Some further observations are reported in the attached file.

Author Response

Response to Reviewer 1 comments:

Thank you a bunch to the esteemed reviewer for their accuracy. Manuscript has been amended as suggested.

  1. We paid attention to the Reviewer’s suggestion and completed the descriptions of our analytical methods (please see paragraphs 3.5, 3.7, 3.8, 3.14 in attached manuscript).
  2. We have made the amendments and all chemicals are listed in paragraph 1 Chemicals (please see lines 740-775 in attached manuscript).
  3. We have corrected the legend inside Figure 2.
  4. We have made the amendments for Table 3 (now Table 4) (please see lines 315-325 in attached manuscript).
  5. We have completed the legend for Table 4 (now Table 5) (please see lines 367-368 in attached manuscript).
  6. We have made the amendments for Figure 4 and we included SI units in the caption (please see line 585 in attached manuscript).
  7. We paid attention to the Reviewer’s suggestion and we included additional information for “Line 556: PBMC, MDA-MB-231 and MCF-7” (please see lines 632-637 in attached manuscript).
  8. We rewritten lines 78-86 (please see lines 93-103 in attached manuscript).

Further comments are included in the revised manuscript and changes are marked in red.

Reviewer 2 Report

Comments and Suggestions for Authors

The article presents a well-structured study that explores the potential of sodium alginate and lecithin-based films enriched with coffee extracts as edible and functional food packaging materials. The research is notable for its comprehensive approach, covering the influence of coffee bean roasting on antioxidant properties, tensile strength, and barrier functions against UV light. The findings, particularly regarding the non-cytotoxic nature of the films and their ability to reduce nitric oxide production, offer promising insights into the safety and health benefits of these materials. In addition, the inhibitory effects of the films on bacterial growth further highlight their potential for food preservation. Overall, this study contributes valuable data to the field of sustainable packaging and offers practical applications without overstating the results. I have some suggestions to improve the manuscript.

 Abstract:

1.  Line 37: should be 0.1 (with point instead of comma, please check whole manuscript if all your data is presented correctly)

2. Lines 34 and 38: add spaces before and after +-

Manuscript:

3.      Table 1: letter d for Green coffee 3-CQA should be in superscript

4.      Add to heading of table 1 that 5-CQA etc. are explained in section 2.3

5. Lines 126 and 129: dm3

6. Line 134: Give 4 numbers after comma for r = 0.99

7.  Table 2: I can’t see names in header raw, I suppose it should be Green coffee, Light roasted etc. …?

8. Paragraph 2.4 and table 2: When you write p-coumaric acid ‘p’ should be written in italics. (check whole manuscript)

9. Paragraph 2.5: cm-1

110. Table 3: add spaces before and after +-

111. Table 3: explain what it means n.m.

112. Table 4: AL water content should be 0.51

113. Line 526: 0.1 ug/mL

114. Paragraph 2.14: add spaces before and after +-

115. Line 554: IC50

116. Line 648: μg·mL-1

117. Although you make a citation of methods from literature, please add paragraph 3.1. Chemicals to Materials and methods listing all chemicals used, their CAS number, purity and provider.

118. Please prepare one Pearson correlation matrix for merged data from tables 1 and 2. Explain observed correlations.

Reviewer 3 Report

Comments and Suggestions for Authors

Introduction: more information devoted to the sodium alginate-based materials used as packaging has to be added.

“This class of compounds is present in significant quantities in green coffee  beans, and in smaller quantities in roasted one Whereas, the second group includes melanoidins, which content increases with increasing degree of roasting, and which are present only in roasted coffee beans.“ lines 107-110 and “During the 111 roasting of green coffee beans, the content of chlorogenic acids decreases, resulting in formation of melanoidins. “ lines 111-112. Please confirm the statements with the relevant references.

dm3  - description of units has to be improved

“the other factors are also involved in creating of antioxidant properties “ line 136. All factors influencing antioxidant properties have to be indicated.

“that during light roasting of green beans the process of thermal decomposition of chlorogenic acids was compensated by formation of melanoidins exhibiting a strong antioxidant properties” lines 158-160. Include relevant references concerning the mentioned statement.

“The presence of CA after hydrolysis was due to hydrolytic decomposition of mono- and dicaffeoylquinic acids. “ lines 171-172. Include relevant references concerning the mentioned statement.

Table 2 is incomplete.

The quality of Figure 1 is unacceptable. Moreover, there is a mistake in the title of this figure.

Figure 1. B and C. Considering that the spectra overlap, the absorbance values should be shown. Moreover, the structure of alginate and lecitin have to be included.

Lines 310. The Authors suggest crosslinking between polymer and phenolic compounds. It has to be proved by the changes in the structure. No new bands can be observed that could confirm this process. Please explain this phenomenon.

The Authors suggest that a crosslinking process can be observed between the polymer and compounds of extracts. The question that arises is why the TS of the new materials significantly decreased.

Mechanical properties. The elongation at break has to be added and discussed.

Section2.6. UV–Vis spectrophotometry and  2.12. Foil barrier properties estimated against UV-Vis light Why have the same properties been discussed twice?

It would be reasonable to assume that the coffee bean extracts are almost brown. Consequently, additional color parameters concerning brown color have to be calculated and discussed.

Figure 4 and Table 1. How is it possible that the total phenolic content and antioxidant activity present higher values for the films than in the case of pure extracts?

“Subsequently, the obtained final solution (alginate 2,5%, w/v; lecithin 5%, w/w and glycerol 1%, w/w) was a control film (AL)” Lines – 682-683. Please explain how it is possible that the solution was a control film.

Line 688 Indicate the volume of solutions, size of Petri dishes, and conditions adopted for drying.

Section 3.9.2. UV–Vis spectrophotometry is ambiguous if the solutions or films are studied.

Conclusions. Please clearly indicate the film characterized by the best properties and justify your choice.

Round 2

Reviewer 3 Report

Comments and Suggestions for Authors

The explanation of the Authors concerning crosslinking between polymer and phenolic compounds should be added to the main text of the publication, not only shown in the response.

“Subsequently, the obtained final solution (alginate 2,5%, w/v; lecithin 5%, w/w and glycerol 1%, w/w) was a control film (AL)” Lines – 798-799. The solution can’t be used as a control sample. The Authors have to add the information that after drying of the solution, the control film was obtained.

In my opinion, the quality of Figure 1 is still not sufficient (subject to Ediotr’s discretion).

All other comments were taken into account, and as a result, the manuscript has been significantly improved.

Author Response

Response to Reviewer comments:

Thank you a bunch to the esteemed reviewer for their accuracy. Manuscript has been amended as suggested.

  1. The explanation of the Authors concerning crosslinking between polymer and phenolic compounds should be added to the main text of the publication, not only shown in the response” - Comments to reviewer: We paid attention to the Reviewer’s suggestion and completed the information (please see lines 354-374 in attached manuscript).
  2. The Authors have to add the information that after drying of the solution, the control film was obtained” - Comments to reviewer: We have made the amendments and completed the information about control sample (please see lines 819-821 in attached manuscript).
  3. In my opinion, the quality of Figure 1 is still not sufficient (subject to Ediotr’s discretion)” - Comments to reviewer: We improved the quality of Figure 1 (1000 dpi) (please see line 246 in the attached manuscript).